# Can LLMs Implicitly Learn Numeric Parameter Constraints in Data Science APIs?

**Yinlin Deng** ⅈ  **Chunqiu Steven Xia** ⅈ  **Zhezhen Cao** ⊛  **Meiziniu Li** ⅊  **Lingming Zhang** ⅈ

ⅈ University of Illinois Urbana-Champaign
⊛ Southern University of Science and Technology
⅊ The Hong Kong University of Science and Technology

{yinlind2,chunqiu2,lingming}@illinois.edu, 12110529@mail.sustech.edu.cn, mlick@cse.ust.hk

## Abstract

Data science (DS) programs, typically built on popular DS libraries (such as Py-Torch and NumPy) with thousands of APIs, serve as the cornerstone for various mission-critical domains such as financial systems, autonomous driving software, and coding assistants. Recently, large language models (LLMs) have been widely applied to generate DS programs across diverse scenarios, such as assisting users for DS programming or detecting critical vulnerabilities in DS frameworks. Such applications have all operated under the assumption, that LLMs can implicitly model the numerical parameter constraints in DS library APIs and produce valid code. However, this assumption has not been rigorously studied in the literature. In this paper, we empirically investigate the proficiency of LLMs to handle these implicit numerical constraints when generating DS programs. We studied 28 widely used APIs from PyTorch and NumPy, and scrutinized the LLMs' generation performance in different levels of granularity: full programs, all parameters, and individual parameters of a single API. We evaluated both state-of-the-art open-source and closed-source models. The results show that LLMs are great at generating simple DS programs, particularly those that follow common patterns seen in training data. However, as we increase the difficulty by providing more complex/unusual inputs, the performance of LLMs drops significantly. We also observe that GPT-4-Turbo can sustain much higher performance overall, but still cannot handle arithmetic API constraints well. In summary, while LLMs exhibit the ability to memorize common patterns of popular DS API usage through massive training, they overall lack genuine comprehension of the underlying numerical constraints.

## 1 Introduction

Data science (DS) is an emerging and important area that combines classic fields like statistics, databases, data mining, and machine learning (ML) to gain insights via complex operations on the abundance of available data [49]. DS libraries (such as PyTorch [41] and NumPy [38]) contain thousands of APIs used by developers and data scientists to process/analyse data. These DS APIs serve as the fundamental building blocks for almost all important ML/DS pipelines, and have penetrated into almost every corner of modern society, including financial systems [18, 4], autonomous driving software [9, 27, 46], coding assistants [45, 37], etc. Due to their high importance and wide usage, automatically synthesizing valid DS programs has been a critical research area [29, 21, 47].

One key challenge of DS code generation is to satisfy the complex constraints within each DS library API. DS library APIs perform transformations (e.g., matrix multiplication) on inputs (i.e., arrays or array-like objects) with numeric constraints on API parameters and inputs. Figure 1 shows an example

38th Conference on Neural Information Processing Systems (NeurIPS 2024).

of a typical *DS program* where the DS library (i.e., PyTorch) is first imported, followed by creating some `input_data`, and then performing the data manipulation operation on the `input_data` using a DS API (`torch.nn.Conv2d`). The parameters of the API (e.g., `kernel_size`, `groups`) must satisfy the corresponding constraints between API parameters and the properties of the `input_data`. We refer to *API constraints* as the set of relationships between properties of `input_data` and API parameters that, if and only if when satisfied, leads to a valid DS API invocation. As seen in Figure 1, not only are there constraints between the properties of the `input_data` and API parameters (e.g., `kernel_size` $\leq$ `H + 2*padding`), but there are also constraints within API parameters (e.g., `out_channel % groups = 0`). These constraints are defined by developers according to the functionality of each DS API, and are usually specified in natural language within the API documentation. Such complex constraints are critical for DS applications, and DS users or even DS experts may unintentionally violate such constraints [29, 26].

Large language models (LLMs) have achieved tremendous success in processing code [10, 2]. Due to their powerful code understanding and generation ability, LLMs have been applied to various coding tasks [34], such as code completion [20, 6], program repair [19, 54], and test generation [16, 17, 48]. For DS libraries, LLMs have been applied to solve practical user queries on StackOverflow [29] and even generate test programs to detect bugs in modern ML frame-

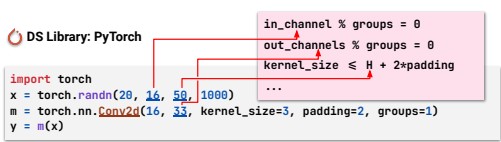

Figure 1: Example DS program with constraints

works [16]. Prior work assumes LLMs, through massive training, can already implicitly model constraints in DS APIs by learning from numerous correct DS API uses [47, 21, 16]. However, this assumption has not been systematically verified. Furthermore, popular DS-specific benchmarks like DS-1000 [29] do not specially test the LLM's ability to satisfy implicit constraints and instead focus on how to apply DS APIs to solve data analysis tasks. These gaps in prior research raise a critical question: *Can LLMs implicitly learn the numeric constraints in data science APIs?*

**Our work.** To answer the question, we conduct a rigorous study on the performance of LLMs in generating valid DS programs satisfying diverse numerical API constraints. We collected a set of 28 representative DS library APIs across two widely-used Python DS libraries (PyTorch and NumPy), each with their unique constraints/setup. Additionally, we categorize each API's constraints into different categories (e.g., equality and arithmetic) and perform in-depth experiments on each constraint type. To support our analysis, we systematically created 3 generation settings: full program, all parameters, and individual parameters, designed to test the LLMs under different evaluation scenarios. Additionally, we vary the difficulty level by adjusting the inputs to explore LLM behaviours when asked to solve more complex API constraints or given more unnatural inputs.

Interestingly, contrary to the popular assumption in prior work, while LLMs can easily satisfy constraints when the inputs are simple, we observe that the performance drops drastically as we increase the difficulty or provide more unusual inputs. We found that LLMs tend to generate simple and common inputs seen during training, highlighting that LLMs are often memorizing patterns instead of truly understanding the actual DS API constraints. For example, for the widely used `Conv2d` API shown in Figure 1, when `max(in_channels,out_channels)` is set to `[128, 256)`, even GPT-4-Turbo [1] can only predict the correct value of `groups` ~24% of the time, while the other models are below 14%. Furthermore, based on our experimental findings, we constructed DSEVAL, the first benchmark for systematically evaluating LLMs' capabilities in understanding the important numerical API constraints for popular DS libraries. DSEVAL contains 19,600 different problems across 12 representative APIs to extensively compare and contrast the performance of different LLMs. DSEVAL supports lightweight and fast evaluation by extracting LLM generated parameters and quickly verifying the correctness using state-of-the-art SMT solvers (such as Z3 [13]) to avoid time-consuming execution-based evaluations. Our evaluation on eight state-of-the-art open-source and closed-source models shows that while all studied models struggle with more difficult problems, GPT-4-Turbo consistently achieves the highest accuracy across all difficulty levels. For example, GPT-4-Turbo achieves an average accuracy of 57.5% for *hard* constraints of PyTorch APIs, while the best open-source model can only achieve 39.2%, demonstrating the huge gap between large proprietary models and other open-source LLMs. Our design of DSEVAL is general and can be easily extended to additional libraries and APIs for the DS domain and even beyond.

Table 1: Categorization of constraint types with exemplar API names, description, and examples.

| Category | API names | Description | Example |
|---|---|---|---|
| Equality | ⬡BatchNorm2d, ⬡Linear
⬡squeeze, ⬡split | Copying specific dimension
Indexing the correct dimension | `nfeat = input_shapes[1]`
`input_shapes[axis] = 1` |
| Inequality | ⬡SoftMax, ⬡mean
⬡sum, ⬡max | Single value related to rank
Multiple values related to rank | $-\text{rank} \leq \text{dim} < \text{rank}$
$-\text{rank} \leq \text{dim} < \text{rank}$ for dim in dims |
| Arithmetic | ⬡MaxPool2d, ⬡AvgPool2d
⬡Conv2d, ⬡Conv1d
⬡reshape, ⬡reshape
⬡Fold, ⬡Conv1d | Multiplies a constant number
Divides a parameter
Product of parameters
Complex arithmetic | `kernel_size` $\leq$ `H + padding * 2`
`in_channels % groups = 0`
$\prod$`input_shapes` $= \prod$`target_shape`
$L=\prod\left\lfloor\frac{\texttt{o\_size[d]}+2\times\texttt{pad[d]}-\texttt{dil[d]}\times(\texttt{k\_size[d]}-1)-1}{\texttt{stride[d]}}+1\right\rfloor$ |
| Set-related | ⬡max, ⬡sum
⬡transpose | Uniqueness
Completeness | `|{dims}| = |dims|`
`{input_shapes} = {axes}` |

## 2 Study Approach

### 2.1 Scope of study

Instead of considering all possible DS programs and APIs, we focus on simple DS programs with only a single API call. This allows us to isolate the evaluation to individual APIs or even individual API parameters, facilitating fine-grained analysis and a detailed examination of the LLMs' limitations with respect to various types of numerical constraint.

We specifically target the core APIs commonly used by users that perform operations on the `input_data`. Additionally, we also only consider *numeric* API constraints: constraints with only numeric parameters such as integers. We ignore any other types of parameters (e.g., string) since they do not affect the validity of numeric constraints. As such, any non-numeric parameters produced by the model will be discarded during constraint validation.

Table 1 shows the types of constraints we considered in the study with the corresponding categories. We group the constraints into *i)* Equality: constraints where the values have to match exactly. We see that equality constraints are related to selecting or generating the right shape in the `input_data`. *ii)* Inequality: constraints where values have to be greater or less than. Inequality constraints include mainly rank related operations to stay within the valid rank range. *iii)* Arithmetic: constraints involving arithmetic operations such as division, modulus or products. There are also more complex API constraints that includes combination of many arithmetic operations. *iv)* Set-related: constraints where the satisfaction criteria depend on different set-based properties. For example, there are constraints that require parameters to be unique or complete with respect to `input_shapes`.

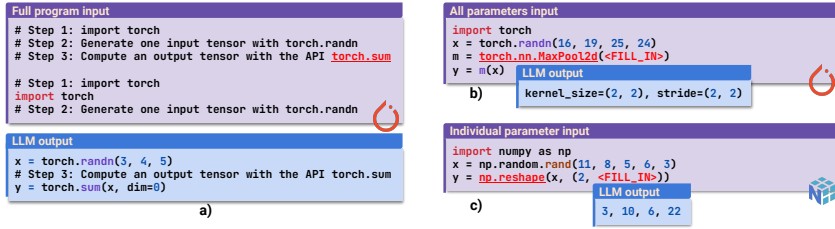

Figure 2: Example problem input and LLM output for each evaluation setting

### 2.2 Evaluation settings

Next, we describe our settings to evaluate the performance of LLM on handling the numeric constraints. In total, we have 3 settings: *i)* full program, *ii)* all parameters, and *iii)* individual parameter.

**Full program.** For the full program setting, we want the LLM to synthesize a complete DS program using a specific API from scratch. To do this, we provide a 3-step instruction and the basic starting code of importing the DS library. Figure 2a shows an example of the full program input for the API `torch.sum` as well as an example LLM output. We note that in this setting, the LLM has full

freedom to generate any type or size for the `input_data`. As such, the LLM may choose very simple `input_data` and API parameter values that can easily satisfy the constraint.

**All parameters.** In the all parameters setting, we directly provide the `input_data` for the API. Figure 2b also shows an example of the input for the API `torch.nn.MaxPool2d` where the LLM just needs to output the API parameters. This setting evaluates if/how LLMs can accurately solve the constraints as we vary the `input_data` with more difficult or uncommon cases. Still the LLM has full freedom to pick the full combination of parameters to satisfy the required constraint.

**Individual parameter (main setting).** To perform a finer-grained evaluation, we introduce the individual parameter setting where we ask the LLM to generate a single parameter of the API. Figure 2c additionally demonstrates an example for `np.reshape` where we only allow the LLM to fill in a single parameter value of `newshape`. Furthermore, we can also add an additional constraint by directly providing the first value of `newshape` (2 in the example). This makes the problem even more challenging where instead of being able to simply copy the `input_shapes`, the LLM now has to reason with the partial shape given and compute the final correct shape to satisfy the constraint. Compared to the prior two settings, the choices here are much limited. This makes the task harder to fully evaluate how LLMs solve complex API constraints, and serves as our main setting.

## 2.3 Input creation and output validation

**Creation.** To produce the inputs for each of the 3 settings, we use a fixed set of templates for each API. For the full program setting, we produce one input per API, changing only the API name in the input instruction. For the all parameters setting, we vary the `input_data` given to the API. In particular, we focus on two properties of the `input_data`: 1) rank of the `input_data` and 2) each dimension value. We create randomized inputs and increase the difficulty by either increasing the rank or the dimension values to measure the LLM performance. Note that input rank or dimensionality can affect different APIs depending on the specific numeric constraints (Table 1). For example, an API like `torch.nn.SoftMax` that has a constraint of `-rank ≤ dim < rank` will have its difficulty influenced by the actual rank of the input tensor. On the other hand, an API like `torch.nn.Conv2d` has a constraint of `in_channels % groups = 0`, which depends on the actual dimension value of the input (i.e., `in_channels`). As the dimension value of `in_channels` increases, it will be more difficult to select the `groups` parameter that can divide it evenly. Therefore, we increase the difficulty of different APIs based on whether the constraint depends on the rank, dimension, or both. Similarly, for the individual parameter setting, we also randomize the `input_data` based on the previous two properties. Additionally, we pick the parameters with interesting constraints for the LLM to predict in order to be representative and cover the major constraint types. Furthermore, since we only ask the LLM to produce a single parameter value, we also vary the other parameter values in the API to add additional constraints (details discussed in Section 4.3).

To ensure the input is valid, we leverage satisfiability modulo theory (SMT) solvers as shown in Figure 3a. SMT solvers, such as Z3 [13], are tools which can be used to solve an SMT problem of determining whether a mathematical or first-order logic formula is satisfiable [5]. We first encode the API constraints into an SMT formula. We then randomly generate *concrete values* for the `input_shapes` and leave the other parameters that we want the LLM to generate as *symbolic variables*. Next, we use an SMT solver to check if the constraints are satisfiable (i.e., there exists a set of values for each symbolic variable that can

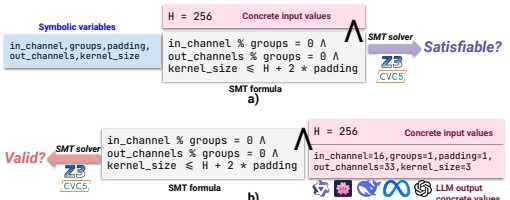

Figure 3: Example usage of constraint solvers to generate inputs and validate outputs.

satisfy the constraint). If it is satisfiable, the input we provide to the LLM is valid, otherwise we restart the process by randomly selecting the concrete values. In our study, we reuse the encoded API constraints provided by NNSmith [32] (a popular tool for testing ML libraries via formal constraint solving) and add additional ones when needed.

**Evaluation.** To evaluate the validity of the DS programs generated by the LLMs, we first parse the output to extract the `input_data` and API parameters. We then check if the LLM predicted

values are valid. This is also done via SMT solving as demonstrated in Figure 3b where we use the SMT formulas and, this time, check if all the concrete values generated are valid according to the constraints. Note that such light-weight constraint solving can support much faster validation than actually executing the generated DS programs, while still providing the same guarantee.

# 3  Experimental Setup

## 3.1  Subjects

We construct a dataset with 28 representative APIs in total from two popular DS libraries: PyTorch (18) and NumPy (10). For our API selection process, we begin by referencing prior work NNSmith [32] and examined all 73 core operators it supports. From these, we select 22 core APIs that have numeric parameter constraints and add additional 6 APIs to obtain the 28 APIs used in our study for both the full program prediction setting (Section 4.1) and the full API parameter prediction setting (Section 4.2). For a more detailed analysis, we select 12 APIs to cover the representative types of numeric constraint for examination in the single API parameter prediction setting (Section 4.3) and in our DSEVAL benchmark (Section 4.4). We use "representative" to mean representative with respect to the numeric parameter constraints in DS library APIs. Table 1 shows the categorization of the different types of numeric constraints that exist in DS libraries. Our selection criteria aim to select a list of APIs that have interesting numeric parameter constraints that can cover all the major constraint categories. A complete list of the 12 APIs and their corresponding constraints is provided in Table 3 in the Appendix.

We focus on the 3 settings described previously to analyse the performance of LLMs. For the full program setting, we generate a single input prompt per each studied API and ask the LLMs to synthesize the complete DS program by varying the sampling temperature. For the all parameters setting, we have 14 difficulty settings, each with 200 different inputs per API, and use greedy decoding to obtain the LLM solutions. The difficulty setting is controlled by increasing the rank of `input_data` (from 2 to 8 in intervals of 1) with default dimension value as `[1,16]`, and increasing the dimension value (i.e., `[1,4]`, `[4,8]`,... , `[128,256]`) with default rank as 3, separately. Finally, in the single parameter setting, we select one parameter for each API for the LLM to generate. For any parameters irrelevant to the constraint, we use the default value if it is an optional parameter, and randomly choose from a reasonable value range if it is a required parameter (Appendix C). We adopt the same difficulty setup and greedy decoding strategy as the all parameter setting.

## 3.2  Metrics

**Validity.** To measure validity, we directly extract the LLM output predictions and evaluate according to the process described in Section 2.3. We define *accuracy* as the percentage of valid programs produced by the LLMs in each difficulty setting.

**Diversity.** To measure diversity, we compute the *unique valid rate*: the percentage of unique valid programs generated via sampling. Note that we deduplicate by extracting the input shapes and numeric parameters, ignoring the irrelevant parameters and irrelevant code suffix.

## 3.3  Studied models.

We evaluate 8 popular state-of-the-art LLMs, including both closed-source and open-source models (detailed list shown in Table 2). For both the full program and all parameter settings, we only present the results for DeepSeek Coder-33b [22], state-of-the-art open-source model, due to the space limit (other models follow similar trends). For the individual parameter setting (the main setting), we focus on the DeepSeek Coder family models (33b, 6.7b, and 1.3b) as well as GPT-4-Turbo (2024-04-09), covering both state-of-the-art open-source and close-source models, as well as models with different sizes. Apart from the full program setting, where the LLM generates a complete program, we perform infilling using the studied LLMs' model-specific infilling format. To perform infilling using GPT-4-Turbo, we design a specialized prompt (see Appendix H). Unless otherwise stated, we use greedy decoding (i.e., temperature = 0) and temperature of 1 when sampling for diversity evaluation.

# 4 Evaluation

## 4.1 Full program prediction

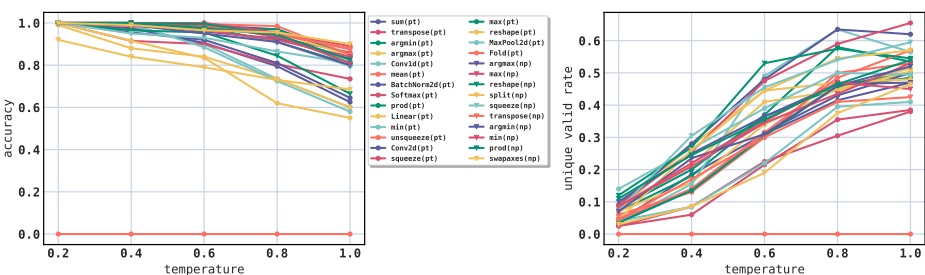

Figure 4: Full program prediction result on all 28 APIs (⟳ PyTorch and 🧊 NumPy).

To start with, we ask the LLM (DeepSeek Coder-33b) to predict the entire DS program from scratch given just simple instructions. Figure 4a shows the overall accuracy of the 18 APIs in PyTorch and 10 APIs in NumPy. We see that with low temperature the model has near perfect accuracy on almost all the APIs and as temperature slowly increases, the accuracy tends to drop (ending with around 0.5~0.8 with temperature=1). Surprisingly, we found that for `torch.nn.Fold`, which contains the most complex constraint, the LLM failed to produce any valid DS programs. This demonstrates that LLMs may still struggle with satisfying the extremely difficult constraints even when given the full freedom of generating any input values. Furthermore, in Figure 4b, we plot the proportion of unique valid programs generated by the model as we vary temperature. Of course when sampling at low temperatures, many of the inputs will be repeated, leading to low number of unique programs in general. In particular, the input shapes are often from widely-used computer vision datasets like `3*224*224` from ImageNet [15]. This indicates the LLMs tend to memorize some common patterns from either documentation or user programs. However, we see that even though the unique valid rate increases with high temperatures to give more diverse and creative outputs, the percentage of unique valid programs can still be mostly below 50%. This demonstrates that while models are successful in generating a high percentage of valid programs, a lot of generated programs are repeated.

## 4.2 Full API parameter prediction

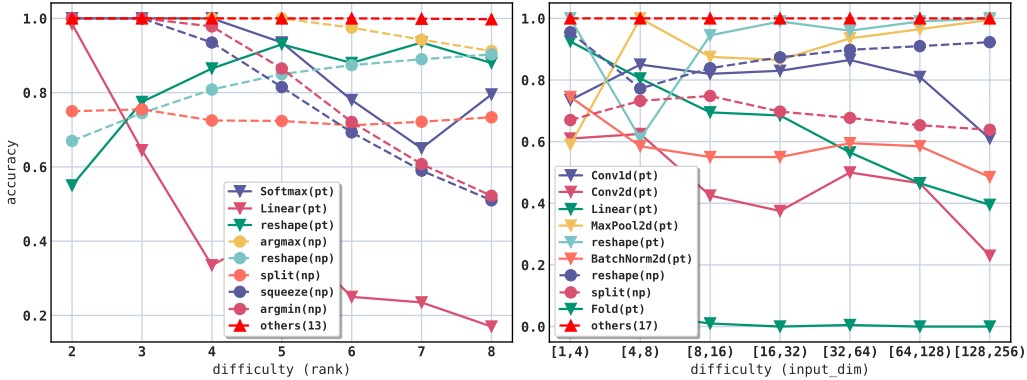

Figure 5: Full API parameter prediction result on all 28 APIs (⟳ PyTorch and 🧊 NumPy). The LLM has near 100% accuracy on some APIs, which are collectively referred to as `others(x)`, where `x` is the number of grouped APIs.

Figure 5 shows the setting where we randomly provide an `input_data` and ask DeepSeek Coder-33b to complete the valid parameters of the API. We vary the difficulty by changing either the rank or the dimension value ranges of the `input_data` to produce more complex and unnatural inputs. We use greedy decoding (temperature 0) to generate one solution per problem, and compute the

average valid rate across the randomly created problems to compute accuracy for each difficulty level. Compared to Section 4.1 where LLMs achieve near-perfect accuracy for almost all APIs with low temperature like 0.2, we observe that the accuracy quickly drops when simply randomizing the input shape, especially for APIs with more complex constraints. This indicates that the learned patterns cannot easily generalize to less common input shapes. We further performed an interesting case study on the PyTorch API `Linear`, and found this phenomenon holds true across different models (Appendix D). However, we see that the majority of APIs maintain high accuracy even as difficulty increases (`others(x)` in Figure 5). This is because these APIs have relatively easy constraints. For example, APIs like `max` or `argmax` only require predicting a single integer representing the dimension to operate on, and the LLMs learn to predict `dim=1` or just rely on the default parameter values of the API which are always valid.

## 4.3   Single API parameter prediction

We now focus on the main finer-grained evaluation setting where we ask LLMs to predict a single parameter value and discuss the input setup, results, and findings for each API separately. Here, we only discuss representative API constraints from each category and full results are in Appendix F.

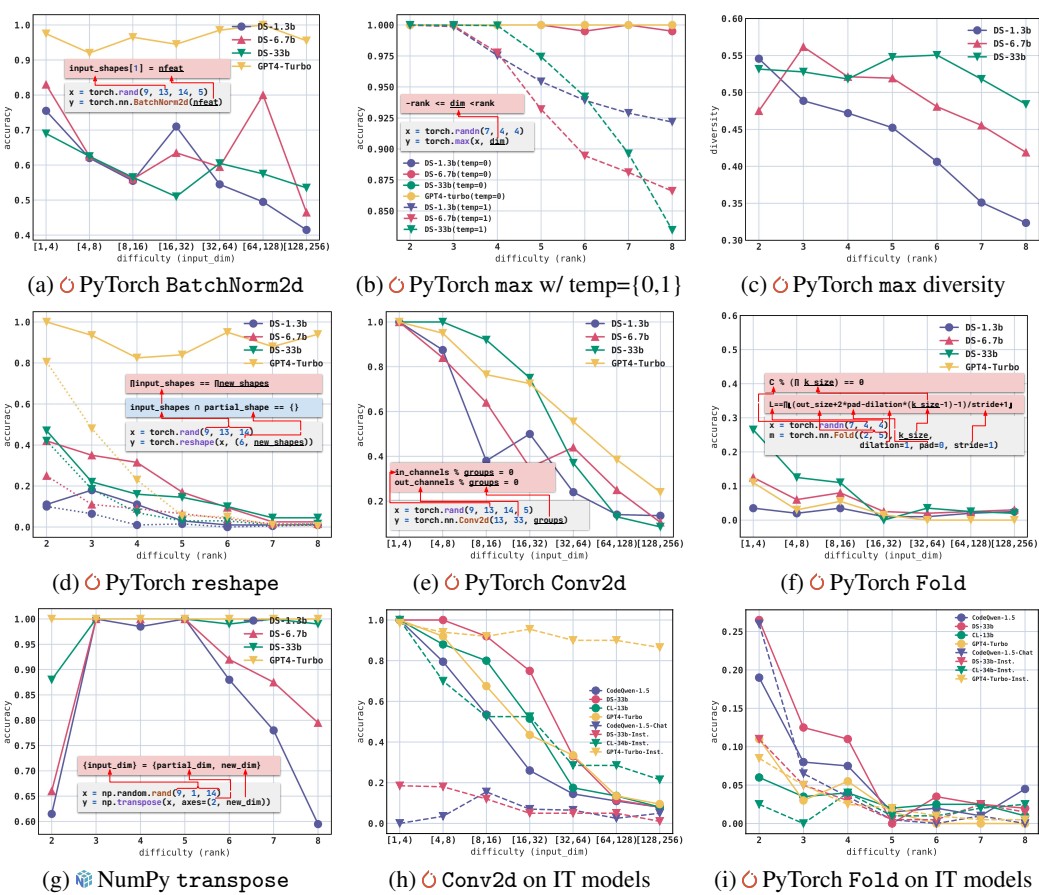

Figure 6: Single API parameter result. Solid lines (except Fig. 6c) show the accuracy of using greedy decoding (`temp=0`). In Fig. 6b, dashed lines show the pass@1 accuracy in sampling experiments with `temp=1`. In Fig. 6d, dotted lines show the accuracy after excluding trivial solutions. In Fig. 6h and 6i, we use *-Inst. to distinguish between the generation settings: infilling (GPT4-Turbo) and free-form generation (GPT4-Turbo-Inst.). More details are provided in Appendix H and I.

**Equality.** `BatchNorm2d` in PyTorch applies batch normalization [25] on a 4D input tensor, with the second dimension as the number of features. We select the parameter `num_features` for the models to predict, with the equality constraint that `num_features = input_shapes[1]`. Figure 6a shows the results as we increase the difficulty by changing the maximum possible value for each input

dimension. We observe that the DeepSeek Coder models drop from around 0.7∼0.8 to less than 0.5, while GPT-4's performance stays around 0.9 throughout different difficulty levels.

***Finding:*** *Overall, we found that smaller LLMs even struggles with even the simple constraint of copying an existing value, while large state-of-the-art LLMs can maintain its high performance.*

**Inequality.** `max` in PyTorch computes the maximum value along a dimension. The parameter we target is `dim` with the valid range being `[-rank, rank)`. In Figure 6b, when using greedy decoding, all 4 LLMs achieve close to perfect accuracy. Therefore, we also conduct sampling experiments and present the pass@1 accuracy and diversity in Figure 6b and 6c. For `max` we compute the diversity differently from Section 3.2 (see Appendix G), since the number of possible unique valid outputs is very small. Interestingly, the smaller DeepSeek Coder-1.3b model achieves highest sampling accuracy for `rank=8`, but has the lowest diversity. This is because the smaller model often predicts common values like `1`, whereas the larger model (33b) can explore various correct answers like `-1,2`.

***Findings:*** *We found that larger models are indeed better at capturing the simple inequality constraints and modeling the true probability of various possible values, while smaller models tend to memorize common patterns, leading to less diverse predictions.*

**Arithmetic.** `reshape` in both PyTorch and NumPy attempts to rearrange the dimensions in the `input_data`, with the constraint being $\prod_i \texttt{input\_shapes}[i] == \prod_j \texttt{new\_shape}[j]$. Since we found that it is common for the LLMs to simply predict the same shape or a permutation of the original, we add an additional constraint: we specify the first dimension of the `new_shape` to be different from any dimensions in `input_shapes`. Figure 6d shows the results as we vary the ranks of the `input_data` for PyTorch (similar trend in NumPy). We observe that most LLMs in the beginning perform well; however, as the difficulty increases, their performance drastically lowers. Meanwhile, GPT-4-Turbo performance does not drop even with more difficult inputs. We found the reason is that GPT-4-Turbo tends to always predict the special `-1` value for `reshape` where the `new_shape` will be automatically inferred by the library. Figure 6d showcases this exact phenomenon in PyTorch (similar trend as NumPy) where dotted lines present the accuracy of any outputs without `-1`. We see that now even GPT-4-Turbo struggles in generating valid parameters without using the `-1` crutch for the constraint.

`Conv2d` in PyTorch applies a 2D convolution over a 4D input tensor. The LLMs are asked to predict the parameter `groups`, where they have to divide both `in_channels` and `out_channels` evenly. The default value for `groups` is the trivial 1 (and therefore always valid). To ensure that there is at least one non-trivial value for `groups`, we randomly sample `in_channels` and `out_channels` within the value range such that their greatest common divisor is greater than 1. Figure 6e shows that the accuracy steadily drops as we increase the magnitude of values: even GPT-4-Turbo can only solve ∼24% of the hardest subset of problems, which other models drop below 14% for the same problems.

`Fold` in PyTorch aims to combine an array of sliding local blocks into a large containing tensor. The constraint required for `fold` is the most complex out of all studied APIs where the LLM tries to generate a `k_size` tuple, and the product of the tuple must divide the 2nd index of the `input_shapes` evenly. Furthermore, it also needs to satisfy a complex equation over multiple parameters as shown in Figure 6f. We use the default values for all parameters other than `out_size` and ask LLMs to produce the correct `k_size`. Shown in Figure 6f, due to the complexity of the constraint, even on the lowest difficulty with small values, LLMs achieve relatively poor accuracy compared to other APIs. As we increase the values, the accuracy drops to nearly 0%. This highlights the high degree of difficulty in many DS APIs which current LLMs cannot reliably solve.

***Findings:*** *Arithmetic parameter constraints in DS APIs are extremely challenging for all LLMs. Our results show that current state-of-the-art LLMs cannot effectively solve such complex constraints with their performance drops drastically and even sometimes drops to zero as we increase the difficulty.*

**Set-related.** `transpose` in NumPy attempts to rearrange/transpose the `input_data` according to the given `new_dim`. In `transpose`, the constraint is that the model-predicted `new_dim` must be a permutation of the original dimensions in `input_data`. We found that the LLMs tend to predict very simple permutations; as such, similar to `reshape`, we directly provide the first dimension of `new_dim` to increase the difficulty. We see that in Figure 6g, LLMs generally perform well on solving this constraint, and their performance improves with larger model sizes. Interestingly, the lowest difficulty of `rank` = 2 has a drop in performance. We theorize that this is because when the `rank` is 2, it is

Table 2: DSEVAL benchmark result. Each column shows both the accuracy/diversity and ranking (🏆).

| | Size | ♻ PyTorch | | | | 🐍 NumPy | | | |
| | | Easy Acc (🏆) | Medium Acc (🏆) | Hard Acc (🏆) | Div (🏆) | Easy Acc (🏆) | Medium Acc (🏆) | Hard Acc (🏆) | Div (🏆) |
|---|---|---|---|---|---|---|---|---|---|
| 🎱 GPT-4-Turbo | NA | 77.2 (1) | 66.2 (1) | 57.5 (1) | - (-) | 95.3 (1) | 85.1 (1) | 71.4 (1) | - (-) |
| 🐋 DeepSeek | 33b | 64.7 (5) | 41.5 (4) | 28.2 (5) | 25.8 (6) | 78.5 (3) | 57.0 (2) | 48.8 (3) | 20.9 (1) |
| | 6.7b | 66.2 (3) | 39.8 (5) | 33.4 (4) | 38.8 (4) | 73.3 (5) | 45.8 (8) | 35.6 (7) | 17.6 (7) |
| | 1.3b | 59.0 (8) | 34.4 (6) | 26.8 (6) | 36.2 (5) | 63.4 (8) | 46.3 (7) | 30.5 (8) | 17.8 (6) |
| 🦙 CodeLlama | 13b | 64.7 (6) | 44.6 (3) | 34.8 (3) | 39.2 (3) | 74.4 (4) | 48.5 (6) | 36.8 (6) | 18.9 (3) |
| | 7b | 62.6 (7) | 32.7 (8) | 13.8 (8) | 21.2 (7) | 67.1 (7) | 53.2 (5) | 45.4 (5) | 18.7 (4) |
| 🌟 StarCoder | 15b | 65.6 (4) | 46.3 (2) | 39.2 (2) | 39.9 (2) | 70.8 (6) | 56.7 (3) | 51.5 (2) | 18.3 (5) |
| 🐳 CodeQwen1.5 | 7b | 67.5 (2) | 33.2 (7) | 25.2 (7) | 53.2 (1) | 80.0 (2) | 54.7 (4) | 47.1 (4) | 19.3 (2) |

more common to directly call `transpose()` without any additional arguments. Therefore, the LLMs struggle a bit when given this unnatural task when asked to predict `new_dim` in low ranks.

***Findings:*** *We found that LLMs generally perform well across the set-related constraints, and their performance scales with increasing model sizes. However, they still struggle with uncommon or unnatural inputs that are no commonly seen during training.*

**Instruction-tuned models.** We additionally investigate the performance of instruction-tuned (IT) LLMs [59] with chain-of-thought (CoT) prompting [51]. Due to computational limitations, we selected 3 constraints from PyTorch on which GPT-4-Turbo (without CoT) performs poorly for this experiment and analysis. The detailed experimental setup is described in Appendix I. Recall that for `Conv2d`, the task is essentially to predict `groups` such that it is a common divisor of two integers. As we observe that some models tend to predict a trivial answer `1`, we specifically mention "Don't set `groups=1`" in the prompt and consider such answer as invalid in evaluation. From Figure 6h, we observe that GPT-4-Turbo with CoT performs well at this non-trivial task, maintaining over 85% accuracy even with values up to 255. By contrast, the best open-source model can only solve 22%! This shows that although models like CodeQwen achieves close performance to GPT-4-Turbo on existing popular benchmarks like HUMANEVAL [10], there is still a huge gap in terms of coding and math reasoning ability between GPT-4-Turbo and other open-source models. Meanwhile, when we use the same setup on the extremely difficult constraint in `Fold`, we see that even GPT-4-Turbo fails to perform well (less than 5% accuracy in later difficulty settings). This demonstrates that while CoT prompting may elicit better performance in constraints like in `Conv2d`, it still cannot effectively handle other more complex arithmetic constraints. In addition to CoT, we also test ReAct [57], another prompting strategy to elicit more reasoning process from LLMs. We observe that while ReAct can perform better than CoT, it still fails to solve more complex arithmetic constraints (detailed in Appendix J). Additionally, we attempt to include API documentation in prompts, but found that this does not always improve performance on our tasks (detailed in Appendix K).

## 4.4 DSEVAL: A public benchmark for numerical DS API constraints

Based on the above findings, we further construct a public benchmark – DSEVAL with the same individual parameter prediction setting and the same representative set of APIs as studied in the Section 4.3. For each API in the benchmark, there are 7 different difficulty settings (grouped as 2 *easy*, 3 *medium*, and 2 *hard* ones) and each with 200 randomly created problems. In total, this gives us 19,600 problems in DSEVAL to extensively evaluate the performance of different LLMs.

Table 2 shows the accuracy and diversity of all 8 models. First, we observe that the LLMs' accuracy drops when increasing the difficulty levels on the benchmark problems. This is also reflected by prior results where LLMs across the board struggle with more difficult problems. Next, we see that GPT-4-Turbo consistently achieves the highest accuracy across all difficulty levels, showing the gap between state-of-the-art proprietary models and other open-source LLMs. Furthermore, we observe some interesting ranking changes across difficulty levels. For example, while CodeQwen1.5 [3] achieves the second-best performance in the lowest difficulty level, its performance drops substantially on the medium and hard problems (second worst on PyTorch medium and hard). Other models like StarCoder [31] improve their relative performance and achieve higher ranking on more difficult

problems, showing that different LLMs can perform differently depending on the input and constraint required to satisfy.

We also study the diversity (see Appendix G for more details) of the LLM outputs, except we do not study GPT-4-Turbo due to its cost. Interestingly, LLMs which achieve high ranking in accuracy do not necessarily perform well in generating diverse correct solutions. This indicates that certain LLMs generate similar solutions to satisfy the constraint, without paying attention to the specific context. Therefore, they are not suitable for tasks like fuzz testing [16] which requires efficiently exploring a large solution space, or for tasks involving uncommon API usage. We further categorize some common mistakes made by LLMs on DSEVAL and provide additional insights in Appendix E. Overall, DSEVAL serves as the first benchmark to systematically evaluate the performance of LLMs on satisfying complex numeric API constraints for popular DS libraries and can be extended to support additional APIs and DS libraries.

## 5   Related work

**LLMs for code.** LLMs have made remarkable advancements in a wide range of coding tasks, including code synthesis [60, 10, 2], debugging [11, 8], repair [53, 54, 7], and analysis [36, 56, 55]. Notably, recent works [29, 16] also demonstrated LLMs' effectiveness in synthesizing DS code, which requires programming proficiency in DS APIs from specialized libraries such as NumPy [38] and PyTorch [41]. Trained on billions of code including such DS code, LLMs, such as StarCoder [31] and DeepSeek Coder [22], have been extensively evaluated on DS code synthesis tasks. However, no prior study has systematically examined whether LLMs can indeed understand numerical API constraints of these scientific libraries instead of just memorizing the trained data [14].

**Coding benchmarks for LLMs.** Most code generation benchmarks [10, 33, 2, 22] are formulated with a natural language description and tests to verify the functional correctness of LLM-generated code. However, these benchmarks mostly target general-purpose code. To access LLM code generation for DS tasks, DS-1000 [29] is created by collecting real DS problems from StackOverflow, and ARCADE [58] evaluates LLMs' ability to solve multiple interrelated problems within DS notebooks. Compared to existing DS benchmarks, our study explores different granularity levels to systematically evaluate to what extent LLMs can implicitly learn DS APIs' numeric parameter constraints.

**Math reasoning of LLMs.** To evaluate LLMs' arithmetic reasoning performance, GSM8K and other benchmarks [12, 42, 35, 24, 28] construct math problems in natural language requiring mathematical computations to solve. Compared to these existing benchmarks, problems designed in our study implicitly encode the arithmetic logic inside the DS library API, and thus can evaluate the LLMs' capability in understanding and solving numerical API constraints in the important DS libraries.

## 6   Conclusion

In this paper, we present the first systematic study on how LLMs understand the numerical API constraints for important DS libraries. Our study results show that current LLMs often memoize common patterns rather than truly understanding the actual numerical API constraints. Moreover, GPT-4-Turbo largely outperforms other open-source models and can well understand some simple arithmetic constraints using CoT. Based on our finding results, we also constructed DSEVAL, the first benchmark (with 19,000 problems) for systematically evaluating LLMs' capabilities in understanding the important numerical API constraints for popular DS libraries (such as PyTorch and NumPy).

## Acknowledgments and Disclosure of Funding

This work was partially supported by NSF grant CCF-2131943 and Kwai Inc. This project is supported, in part, by funding from Two Sigma Investments, LP. Any opinions, findings, and conclusions or recommendations expressed in this material are those of the authors and do not necessarily reflect the views of Two Sigma Investments, LP.

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

# A  Problem statement

We first begin by describing the type of programs we are targeting as well as any terminology definitions. In our study, we aim to verify the ability of LLMs on satisfying the numerical constraints for DS library APIs. Figure 1 shows an example of a typical *DS program* consisting of different DS APIs created by following these steps: *i)* importing the DS library (e.g., PyTorch) to be used in the program; *ii)* obtaining or generating input data; *iii)* performing data manipulation operation using the input data to produce outputs. In order for a DS program to be valid, it needs to satisfy the constraints required in the DS library APIs used. Next, we will describe each component of a DS program in more detail.

**DS APIs.** After importing the DS library, we start by obtaining or creating some input data to be used in the program. The data creation process is done using a specific type of DS APIs, refer to by us as *generation* APIs. The output of generation APIs is a specific data structure (e.g., tensors, arrays, dataframes) used by each DS library defined by the parameters of the APIs. In the example from Figure 1, the generation API is `torch.randn` to produce the input data of `x`. The DS library specific `input_data` commonly has the following properties: *i)* `shape`: the size, rank or dimensions of the data structure (e.g., [20, 16, 50, 1000]) *ii)* `dtype`: the type of primitive data in the `input_data` (e.g., `float`) *iii)* `value`: the exact data values in the `input_data`. While most of the constraints in DS library APIs focus on relationship of the `shape`, both `dtype` as well as `value` can be important to satisfy additional constraints. Furthermore, more complex generation APIs can create heterogeneous `input_data` that contains different `dtypes` or even nested structures.

Using the `input_data` created by the generation API, DS programs then use *manipulation* APIs to perform additional operations. Different from generation APIs where the output produced depends only on the provided API parameters, manipulation APIs create the output based on both the API parameters and the input data. We refer to *API parameters* as the options used to initialize the behaviour of the *manipulation* API. In the example, the *manipulation* API is `Conv2d` and is first defined with a set of parameters (e.g., `kernel_size`) and then in the next line applied on `x` to create the output `y`. For the program to be valid, the parameters of the manipulation API must satisfy the corresponding constraints between API parameters and the properties of the `input_data`. Note that we consider both model manipulation APIs (like `Conv2d` in the example) as well as sequential manipulation APIs where the `input_data` is directly provided as a parameter of the API.

**API constraints.** We refer to *API constraints* as the set of relationships between properties of `input_data` and API parameters that, if and only if when satisfied, leads to a valid DS API invocation. Figure 1 provides some of the example *API constraints* for `Conv2d`. We observe that not only are there constraints between the properties of the `input_data` and API parameters (e.g., `in_channel % groups = 0`, `kernel_size ≤ H + 2 * padding`), but there are also constraints within API parameters (e.g., `out_channel % groups = 0`). Failure to satisfy any one of those constraints will lead to an invalid DS API innovation where, when executed by the DS program, will lead to a runtime error.

# B  Benchmark details

Table 3 lists all the 12 APIs we studied in Section 4.3 and Section 4.4, where we ask LLMs to predict a single API parameter. In Column "Constraint", the underlined parameter is the one that the models need to predict, and we also list all the parameter constraints related to it. Column "Category" presents the categorization following Table 1, highlighting that our selected APIs and API parameters can cover all the categories and are representative of their group.

**Difficulty settings.** As discussed in Section 3.1, we have two different difficulty settings depending on the specific APIs, namely `rank` and `rank`. Below are the detailed setups for each API.

- For `torch.max`, `np.squeeze`, `np.argmax`, `np.transpose`, `np.max` whose constraints that are more related to tensor dimensions, we design the difficulty level by increasing the rank of input data (i.e., how many dimensions it has).

- For `torch.nn.BatchNorm2d`, `torch.nn.Fold`, `torch.nn.Conv2d`, `torch.nn.MaxPool2d`, `np.split` whose constraints that are more related to the actual values (either API parameter or the dimensions of input data), we design the difficulty level by increasing the range of relevant values.

- For `torch.reshape`, `np.reshape`, since their constraints are closely related to both rank and value, we study both settings for each of them.

Table 3: List of APIs and corresponding constraints used in DSEVAL.

| Library | API full name | Constraint | Category |
|---|---|---|---|
| ⟳ PyTorch | `torch.nn.BatchNorm2d` | `num_features` = `input_shape[1]` | Equality |
| ⟳ PyTorch | `torch.max` | `-rank` $\leq$ `dim` < `rank` | Inequality |
| ⟳ PyTorch | `torch.nn.Fold` | `L` = $\prod \lfloor \frac{\text{o\_size}[d] + 2 \times \text{pad}[d] - \text{dil}[d] \times (\text{k\_size}[d]-1)-1}{\text{stride}[d]} +1 \rfloor$ $\wedge$
`C` % $\prod$ `k_size` = 0 | Arithmetic |
| ⟳ PyTorch | `torch.nn.Conv2d` | `in_channels` % `groups` = 0 $\wedge$
`out_channels` % `groups` = 0 | Arithmetic |
| ⟳ PyTorch | `torch.nn.MaxPool2d` | `kernel_size` $\leq$ `H` + 2 $\times$ `padding` | Arithmetic |
| ⟳ PyTorch | `torch.reshape` | $\prod$`input_shape` = $\prod$`target_shape` | Arithmetic |
| 🧊 NumPy | `np.squeeze` | `input_shape[axis]` = 1 | Equality |
| 🧊 NumPy | `np.argmax` | `-rank` $\leq$ `axis` < `rank` | Inequality |
| 🧊 NumPy | `np.reshape` | $\prod$`input_shape` = $\prod$`target_shape` | Arithmetic |
| 🧊 NumPy | `np.split` | `input_shape[axis]` % `section`=0 | Arithmetic |
| 🧊 NumPy | `np.transpose` | `-rank` $\leq$ `dim` < `rank` for dim in `axes` $\wedge$
{`input_shapes`} = {`axes`} | Inequality
Set-related |
| 🧊 NumPy | `np.max` | `-rank` $\leq$ `dim` < `rank` for dim in `axis` $\wedge$
\|{`axis`}\| = \|`axis`\| | Inequality
Set-related |

## C   Common parameter value ranges

To create a set of diverse problems for LLMs to predict a single API parameter, we randomize the context, namely the input data shape and other API parameters. Since the goal of our study is to focus on numeric API constraints, we want to control the complexity or naturalness of the constraint-related variables, and ensure that the other unrelated parameters are always within a reasonable and common range. More specifically, during problem creation, if the irrelevant API parameter is an optional parameter, we just use its default value. If the irrelevant API parameter is a required API parameter, then we randomly choose a value according to the common value range listed in Table 4. For example, for `torch.nn.Conv2d(in_channels, out_channels, kernel_size, groups=1)`, the targeted parameter is `groups`, and `kernel_size` is a irrelevent but required parameter. Therefore, across all difficulty levels, we pick `kernel_size` randomly from `[1,10]`.

To obtain Table 4, we perform an extensive study and refer to developer-written unit tests [44, 40], API documentations [43, 39], and existing DS library fuzzing literature [50, 23, 32, 30] to gather the common value range for each API parameter.

## D   Case study of the Linear API

When evaluated using the full API parameter setting (Section 4.2), we observe a much more significant accuracy drop for `Linear` compared to the other APIs (Figure 5). To gain deeper insights, we conduct an in-depth case study of this API.

For `torch.nn.Linear(in_features, out_features)`, the only constraint is that the `in_features` should match the last dimension of the input tensor. However, the DeepSeek Coder-33b model tends to copy the wrong dimension of the input tensor, likely because it has not seen lots of high-rank tensors (rank > 4) in the pre-training data.

Table 4: List of APIs and corresponding common input range used in DSEVAL.

| API full name | Parameter name | Range |
|---|---|---|
| Conv[1\|2]d (⟳) | in_channels | [0, 128] |
|  | out_channels | [0, 128] |
|  | kernel_size | [1, 10] |
|  | stride | [1, 5] |
|  | padding | [0, 9] |
|  | dilation | [1, 5] |
| MaxPool2d (⟳) | kernel_size | [1, 10] |
|  | stride | [0, 5] |
|  | padding | [0, 9] |
| BatchNorm2d (⟳) | num_features | [0, 256] |
| expand (⟳) | last_dim | [-10, 10] |
| [argmin\|argmax] (⟳,▨) | keepdims | [True, False] |
| reshape (⟳,▨) | out_shape | [0, 256] |
| Linear (⟳) | in_features | [1, 256] |
|  | out_features | [1, 256] |
|  | bias | [True, False] |
| Fold (⟳) | output_size | [2, 10] |
|  | kernel_size | [1, 10] |
|  | stride | [1, 5] |
|  | padding | [0, 9] |
|  | dilation | [1, 5] |
| APIs' input_data (⟳,▨) | input_shape | [0, 256] |
|  | input_rank | [2, 8] |

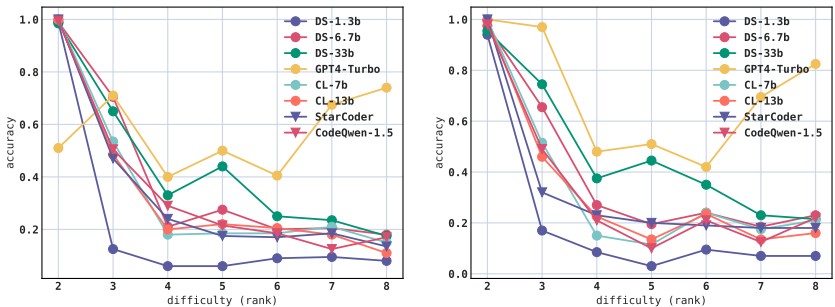

(a) ⟳ PyTorch Linear Full API parameter   (b) ⟳ PyTorch Linear Single API parameter

Figure 7: Result on `torch.nn.Linear` using 8 different LLMs

We further evaluate this phenomenon across different LLMs. Figure 7b shows the results for both the full API parameter and single API parameter setting for `torch.nn.Linear` as we increase the difficulty (rank of the input data) across 8 LLMs. Similar to DeepSeek Coder-33b, the performance of other LLMs also drops significantly when rank reaches 4. Afterwards, the performance stabilizes for higher difficulties (i.e., rank > 4) especially for open-source LLMs. This is true for both the full API parameter and single API parameter setting.

Surprisingly, we found that even the state-of-the-art GPT-4-Turbo drops in performance when the rank reaches 4. However, we see that GPT-4-Turbo was able to improve its performance in higher difficulties (i.e., rank > 6). After looking at the results, we found that for lower ranks, GPT-4-Turbo

tends to use other APIs as "short-cuts" and forgo the analysis on `torch.nn.Linear` directly as shown in Figure 8.

```
import torch
x = torch.randn(1, 8, 10, 10)
m = torch.nn.Linear(8*10*10, 3)
y = m(x.view(1, -1))
```

Figure 8: Example of GPT-4-Turbo's incorrect response. In this example, instead of operating on the original input tensor x and set `in_feature` to its last input dimension (10), GPT-4-Turbo multiplies all the dimensions together and performs a flattening operation (`x.view(1, -1)` before invoking `Linear`). This violates the instruction to the model, which prohibits modifying the API invocation code. As such, this model response is evaluated as incorrect.

## E  Common mistakes made by LLMs

In this section, we categorize some common mistakes made by LLMs during our experiments and offer some additional insights and explanations.

- **LLMs struggle with uncommon input tensors:** We found that across many APIs and constraints, LLMs struggle when provided with uncommon input tensor ranks (i.e., rank > 4) or uncommon shapes (e.g., `x = torch.rand(9, 30, 23, 4)`). The reason is that LLMs are mostly trained with data that contains very common shapes or ranks. As such, LLMs can easily make mistakes on uncommon inputs.
- **LLMs tend to predict common parameter values blindly:** We also observe that LLMs tend to generate common parameter values (e.g., 0, 1, powers of 2) which often turn out to be incorrect. This is again because LLMs are trained with pre-training code that frequently contains such parameter patterns and thus are likely to predict them even given a different input context.
- **LLMs pay attention to the wrong tokens/irrelevant parameters:** LLMs can learn spurious correlations and pay attention to the wrong context tokens. For example, open-source LLMs struggle with the simple equality constraint `in_features=input.shape[-1]` in `torch.nn.Linear` because the attention weights are focused on the irrelevant parameters.

## F  Additional individual API parameter results

In this section, we provide the additional results for the individual API parameter setting.

**Reshape.** `reshape` in NumPy contains the same functionality and constraint as the PyTorch. Figure 9a shows the result across two difficulty dimensions. Similar to the result discussed in Section 4.3, we also observe the same trend in NumPy where GPT-4 is able to achieve superior performance.

**MaxPool2d.** `MaxPool2d` in PyTorch applies a 2D max pooling over a 4D input tensor. We focus on predicting the API parameter `padding`, where it needs to satisfy the following constraint: `kernel_size` $\leq$ `H + 2*padding` (the problem is already simplified by setting `stride` to 1 and setting `W=H`). Figure 9b shows that even GPT-4 is incapable of getting this type of non-trivial linear constraints right when the value range increases to $[128, 256]$, and we observe that it tends to predict 0 which is the default value for `padding` and therefore fails to satisfy the second constraint. Meanwhile, DeepSeek Coder models, especially the 6.7B variant, do surprisingly well in the highest difficulty level we studied. After inspecting the outputs, we find that the DeepSeek Coder-6.7B model is able to predict an expression `kernel_size//2` instead of a constant number for `padding`, and since the expression correctly characterizes the constraints it always leads to a valid solution.

**Squeeze.** `squeeze` in NumPy aims to remove any dimension of length one from the `input_data`. The constraint is for the LLM to predict a dimension `dim` where $\text{input\_shapes}[\text{dim}] = 1$. We add an additional constraint when generating the `input_data` such that there is only one dimension with length one (only one correct dimension). Figure 9c shows the result as we increase the rank

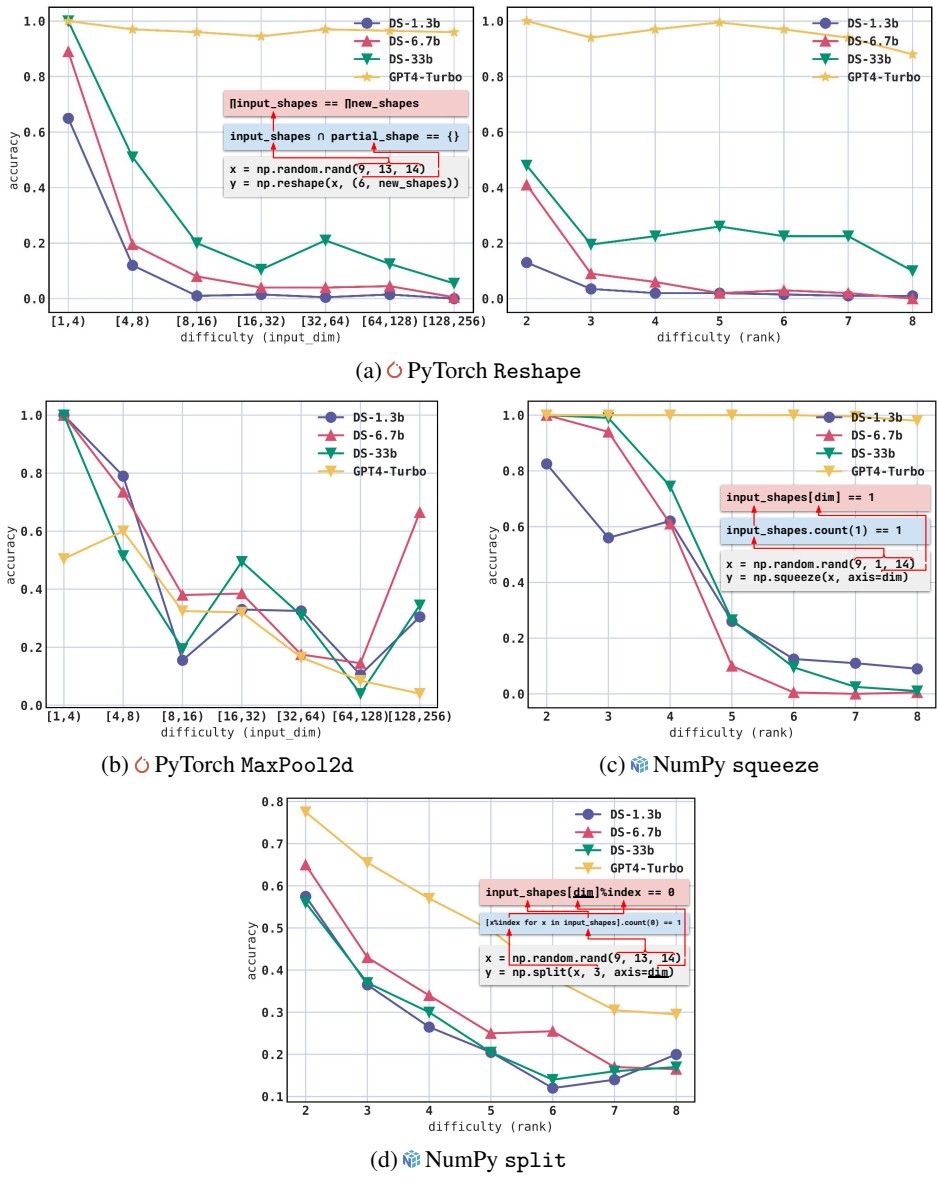

(a) ⟳ PyTorch `Reshape`

(b) ⟳ PyTorch `MaxPool2d`

(c) 🧊 NumPy `squeeze`

(d) 🧊 NumPy `split`

Figure 9: Single API parameter result (cont).

in `input_data`. We observe that while in smaller ranks (2-3), the LLMs achieve close to perfect accuracy, the performance quickly drops off when we increase the rank to be >4, where all 3 LLM sizes perform similarly. We found that this is because LLMs tends to generate dimensions of 0 or 1 (commonly seen in example code and pre-training data). As such, when given a high ranked tensor where the correct squeeze dimension can be much higher than 0 or 1, the LLM struggles to satisfy the simple constraint. This again demonstrate the memorization issue where LLMs tend to predict commonly seen parameters during training instead of reasoning over the actual API constraint.

`Split.` For `split` in NumPy, the goal is to predict a dimension `dim` which can divide the index parameter `index` evenly. Just like `squeeze`, we also add an additional constraint when generating the `input_data` such that there is only one dimension that is divisible by `index`. Figure 9d shows the results, where we see that even with rank of 2, the accuracy is just above 50%, showing the increase in difficulty of the constraint in `split` As the rank increases, we also observe a huge decrease in performance, where LLMs again overwhelmingly predict dimensions of 0 or 1.

# G   Diversity metric

Besides accuracy (percentage of valid programs generated), we also measure the *diversity* of the valid programs generated. In particular, based on the exact API, we use different diversity metric:

i) For APIs where the number of possible valid outputs are large and not restricted (e.g., `torch.reshape`), we sample the LLM multiple times (100) with high temperature (1) and compute the percentage of unique valid programs generated as discussed in Section 3.2.

ii) For APIs where the number of possible valid outputs are fixed (e.g., `np.max` can only select valid dimensions within a range), we again sample the LLM multiple times and then compute the distance between uniformed valid distribution and the distribution produced by the LLM. For example, if the set of all valid answers is {-1,0,1}, and the model predicts {-3,-2,0,0,0,1,1,1,1,1} in 10 samples, then the LLM's distribution is `P={-1: 0, 0: 0.3, 1: 0.5, others: 0.2}`, and the reference distribution is `Q={-1: 1/3, 0: 1/3, 1: 1/3, others: 0}`. Next, we compute the Hellinger distance [52] between the two discrete probability distributions and compute the diversity as $1 - distance$.

Since it requires a large amount of sampling programs (100 samples per problem) to evaluate diversity, in Section 4.4, we evaluated the diversity of each LLM only on a single difficulty level, i.e., the third level, either `rank=4` or `value in [8,16]`.

# H   GPT-4-Turbo infilling prompt

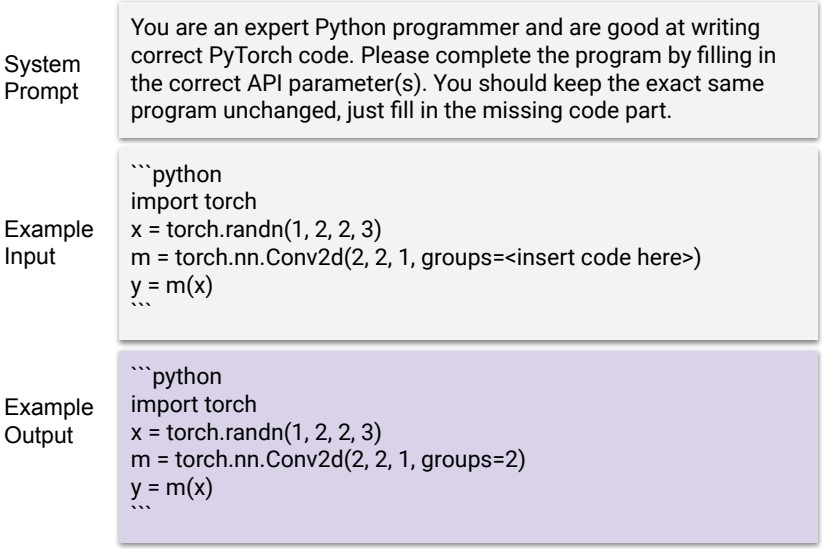

Figure 10: Example GPT-4-Turbo prompt used for infilling and output

GPT-4-Turbo is an instruction-following LLM and we do not have access to a base version that supports direct infilling. Therefore, we use a infill-specific prompt to ask GPT-4-Turbo to only fill in the missing code without adding any additional text. This setup allows us to compare against other infilling LLMs in the same setting. On the other hand, in this paper (e.g., Fig. 6h), we use "GPT-4-Turbo-Inst" to differentiate the free-form generation setting. "GPT-4-Turbo-Inst" indicates that we allow it to generate additional text (such as CoT or ReAct reasoning steps).

Figure 10 shows the prompt used by us to perform infilling using GPT-4. Note we separate out the system prompt and how we format an example input. Additionally, we modify the library name in the system prompt depending library of the API.

Figure 11: CoT prompts used for the instruction LLMs

## I  Single API parameter results for instruction model with CoT prompting

**Experiment setup.**  We design CoT prompts shown in Figure 11. Note that the prompt is slightly different for different models due to their specific format requirements. Additionally, for `torch.nn.Conv2d`, we use a custom prompt (shown in Figure 12) where we explicitly add a sentence "Don't set groups=1" to avoid trivial answers. We use greedy decoding and set `max_new_tokens` to 512 for all models and all APIs, except for `torch.nn.Fold` we use `max_new_tokens=1024` since LLMs tend to predict longer text before the actual code for this challenging API.

**Results.**  Figure 13 shows the additional results on constraints chosen to evaluate the instruction-tuned LLMs. Figure 13a, 13b, and 13c shows the result of the different variants on `Conv2d`. Furthermore, we also evaluate on the constraint in `MaxPool2d` with the result shown in Figure 13d.

## J  Single API parameter results for instruction model with ReAct prompting

We applied the ReAct prompting strategy [57] on the difficult APIs constraints studied using CoT in Section 4.3. Figure 14 shows our ReAcT prompt for GPT-4-Turbo. Note that the prompt is slightly

**GPT-4-Turbo CoT Prompt for Conv2d.**

You are an expert Python programmer and are good at writing correct {library} code. Please complete the program by filling in the correct API parameter(s). Don't set groups=1.

**Example Input**

```python
import torch
x = torch.randn(254, 180, 153, 189)
m = torch.nn.Conv2d(180, 135, 8, groups=<insert code here>)
y = m(x)
```

**Example Output**

In the given code snippet, you are trying to use a `torch.nn.Conv2d` layer, which is a 2D convolutional layer in PyTorch. The parameters for `torch.nn.Conv2d` are:

1. `in_channels` (int) – Number of channels in the input image.
2. `out_channels` (int) – Number of channels produced by the convolution.
3. `kernel_size` (int or tuple) – Size of the convolving kernel.
4. `stride` (int or tuple, optional) – Stride of the convolution. Default: 1
5. `padding` (int or tuple, optional) – Zero-padding added to both sides of the input. Default: 0
6. `dilation` (int or tuple, optional) – Spacing between kernel elements. Default: 1
7. `groups` (int, optional) – Number of blocked connections from input channels to output channels. Default: 1
8. `bias` (bool, optional) – If `True`, adds a learnable bias to the output. Default: True

For the `groups` parameter in your code, you need to specify an integer that divides both the number of input channels (`in_channels`) and the number of output channels (`out_channels`). The `groups` parameter is used to control the connections between inputs and outputs. Setting `groups` to 1 means all inputs are convolved to all outputs. If `groups` is set to `in_channels`, it means each input channel is convolved with its own set of filters (like a depthwise convolution).

In your case, since `in_channels` is 180 and `out_channels` is 135, you need to find a divisor of both 180 and 135 to use as the `groups` value. The greatest common divisor (GCD) of 180 and 135 is 45. Thus, you can set `groups=45`:

```python
import torch
x = torch.randn(254, 180, 153, 189)
m = torch.nn.Conv2d(180, 135, 8, groups=45)
y = m(x)
```

This configuration means that the convolution operation is performed separately in 45 groups, each group handling a subset of the input and output channels.

Figure 12: CoT prompt and example input/output for `torch.nn.Conv2d` used for the instruction LLMs. In this example, GPT-4-Turbo is able to accurately recall the numeric constraint for the targeted parameter `groups`, apply them on this specific input example (`in_channels=180, out_channels=135`), and generate valid answer 45.

different for different models due to their specific format requirements. Our ReAct prompt asks the LLM to first generate a thought and then produce the code output. We also provide the LLM with a single demonstration of the ReAct task. Furthermore, similar to the CoT experiments, we also explicitly add the sentence "`Don't set groups=1`" to avoid trivial answers.

Figure 15 compares the results of the instruction-following LLMs using CoT versus ReAct as well as their base variants. We see that for `torch.nn.MaxPool2d`, ReAct prompting generally performs

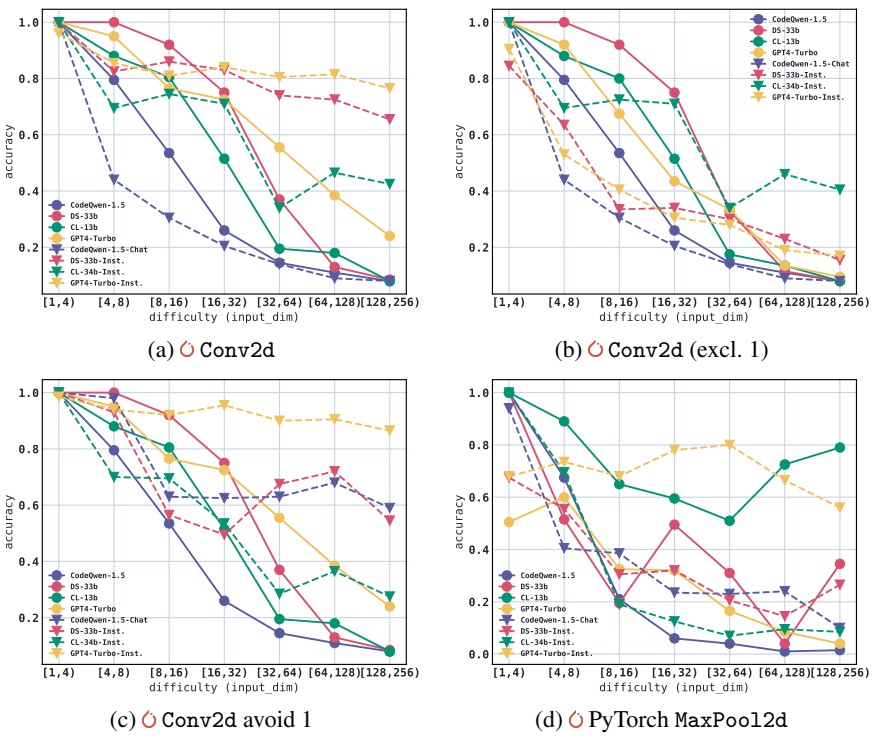

(a) ⟳ `Conv2d`

(b) ⟳ `Conv2d (excl. 1)`

(c) ⟳ `Conv2d avoid 1`

(d) ⟳ PyTorch `MaxPool2d`

Figure 13: Instruction model results. Figure 13a shows the result where we do not add the additional non-trivial requirement in the prompt ("Don't set groups=1"), and we also count `groups=1` answers as correct. Figure 13b shows the result on the same setting and model samples as Figure 13a but we count `groups=1` answers as incorrect. Figure 13c shows the result where we add the additional non-trivial requirement in the prompt ("Don't set groups=1"), but we still count `groups=1` answers as correct.

better than CoT especially in more difficult problem settings (e.g., at highest difficulty setting, GPT-4-Turbo-ReAct: 89.5% versus GPT-4-Turbo-CoT: 56.0%). This demonstrates the effectiveness of ReAct in generating thoughts that can help with the correct API parameter generation. However, for `torch.nn.Conv2d`, ReAct performs similarly to CoT prompting. The reason is that the constraint used in `Conv2d` is much more complex, requiring factorization. As such, smaller open-source LLMs cannot perform well even with reasoning steps. On the other hand, state-of-the-art LLMs like GPT-4-Turbo show their powerful reasoning abilities by improving the performance over the base variant with both CoT and ReAct. Although ReAct performs better than CoT for the easier difficulty settings in `torch.nn.Fold`, its performance quickly drops in higher difficulty settings (at best ∼5% accuracy with the best GPT-4-Turbo). Overall, this experiment results demonstrate that even more advanced prompting methods such as ReAct still cannot effectively handle more complex constraints.

## K    Single API parameter results with documentation-augmented prompting

We conducted additional experiments using the documentation-augmented setting across the 3 difficult API constraints used in the CoT experiments (Section 4.3). We provide the raw documentation of each API (obtained from the source code docstring) in the prompt and apply both base and instruction-following LLMs. Figure 16 shows an example prompt to perform the documentation-augmented setting for GPT-4-Turbo. Note that the prompt is slightly different for different models due to their specific format requirements. Furthermore, similar to the CoT and ReAct experiments, we also explicitly add the sentence "Don't set groups=1" to avoid trivial answers.

In Figure 17, we compare the performance with and without documentation. We found that there are cases where documentation can improve performance. For example, in the most difficult setting of `torch.nn.Conv2d`, adding documentation is able to improve performance of CodeLlama-34b-

Figure 14: Example React prompt used for GPT-4-Turbo

Instruct from 20% to 45% accuracy (Figure 17d). However, there are also similar cases where adding documentation decreases performance. For example the GPT-4-Turbo-Instruct performance falls from 57.5% to 22.5 in the most difficult setting of `torch.nn.MaxPool2d` (Figure 17b).

Since we provide the raw documentation text without further processing, the success rate of adding documentation can vary depending on the specific model as well as the quality of the documentation. As such, this demonstrates that naively adding API documentation cannot always achieve better performance on our tasks.

## L   Computation Environment

We perform both LLM generation and evaluation on an 64-core workstation with 256 GB RAM running Ubuntu 20.04.5 LTS. For local open-source LLMs, we use NVIDIA RTX A6000 GPUs. For GPT-4-Turbo experiments, we directly access the API endpoint provided by OpenAI.

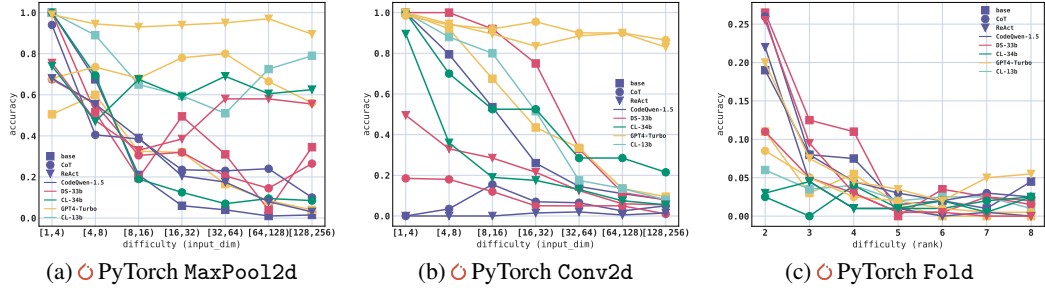

(a) ⟳ PyTorch `MaxPool2d`  (b) ⟳ PyTorch `Conv2d`  (c) ⟳ PyTorch `Fold`

Figure 15: Single API parameter result with chain-of-though (CoT) and ReAct prompting and the base LLM using greedy decoding with 200 problems for each difficulty setting. We follow the same generation and evaluation setting used in Section 4.3.

You are an expert Python programmer and are good at writing correct PyTorch code. Please refer to the given API documentation and complete the Python program. Documentation for the torch.nn.MaxPool2d API:
Applies a 2D max pooling over an input signal composed of several input planes.

In the simplest case, the output value of the layer with input size :math:`(N, C, H, W)`,
output :math:`(N, C, H_{out}, W_{out})` and :attr:`kernel_size` :math:`(kH, kW)`
can be precisely described as:

.. math::
    \begin{aligned}
        out(N_i, C_j, h, w) ={} & \max_{m=0, \ldots, kH-1} \max_{n=0, \ldots, kW-1} \\
                        & \text{input}(N_i, C_j, \text{stride[0]} \times h + m,
                                    \text{stride[1]} \times w + n)
    \end{aligned}

If :attr:`padding` is non-zero, then the input is implicitly padded with negative infinity on both sides
for :attr:`padding` number of points. :attr:`dilation` controls the spacing between the kernel points.
It is harder to describe, but this `link`_ has a nice visualization of what :attr:`dilation` does.

Note:
    When ceil_mode=True, sliding windows are allowed to go off-bounds if they start within the left padding
    or the input. Sliding windows that would start in the right padded region are ignored.

The parameters :attr:`kernel_size`, :attr:`stride`, :attr:`padding`, :attr:`dilation` can either be:

    - a single ``int`` -- in which case the same value is used for the height and width dimension
    - a ``tuple`` of two ints -- in which case, the first `int` is used for the height dimension,
      and the second `int` for the width dimension

Args:
    kernel_size: the size of the window to take a max over
    stride: the stride of the window. Default value is :attr:`kernel_size`
    padding: Implicit negative infinity padding to be added on both sides
    dilation: a parameter that controls the stride of elements in the window
    return_indices: if ``True``, will return the max indices along with the outputs.
            Useful for :class:`torch.nn.MaxUnpool2d` later
    ceil_mode: when True, will use `ceil` instead of `floor` to compute the output shape

Please complete the program by filling in the correct API parameter(s). You should keep the exact same program unchanged, just fill in the missing code part.

GPT-4-Turbo Documentation-augmented Prompt

Figure 16: Example documentation-augmented prompt used for GPT-4-Turbo

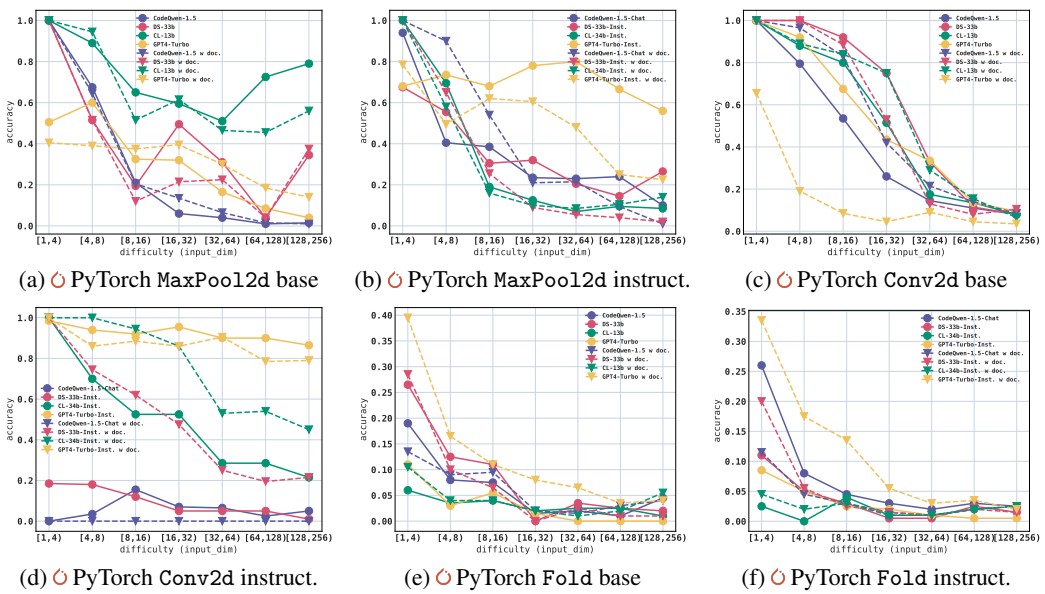

Figure 17: Single API parameter result for both instruction-following and base LLMs with and without documentation. We follow the same generation and evaluation setting used in Section 4.3.

