# OpenReview forum: "Can LLMs Implicitly Learn Numeric Parameter Constraints in Data Science APIs?"
_NeurIPS.cc/2024/Conference — NeurIPS 2024 poster_

### Official Review · Reviewer_WbTY · 2024-07-03

**Soundness:** 3
**Presentation:** 3
**Contribution:** 2
**Rating:** 6
**Confidence:** 4

**Summary:**

In data science library APIs, there is a numerical parameter constraint between input data and parameters. This paper presents an empirical study on whether large language models (LLMs) can implicitly learn numerical constraints in data science library APIs. The findings indicate that LLMs demonstrate the ability to memorize common patterns in the use of popular data science APIs, and their performance significantly declines when the rank or dimensions of the input data increase, suggesting a general lack of true understanding of the API's numerical constraints. Additionally, the paper reports several other findings. It also introduces a dataset named DSEVAL, designed to evaluate LLMs' capabilities in understanding numerical API constraints for some popular data science libraries.

**Strengths:**

1. The paper explores a new and interesting problem regarding whether LLMs comprehend the numerical parameter constraints between input data and parameters in Data Science library APIs during the code generation process.

2. The study is thoroughly conducted, encompassing three scenarios (Full program, all parameters, Individual parameter) and various series, different sizes of LLMs such as GPT-4-Turbo, DeepSeek, CodeLlama, StarCoder, and CodeQwen 1.5.

3. It contributes a new dataset designed to assess LLMs’ capabilities in understanding the numerical API constraints for popular Data Science libraries like PyTorch and NumPy, with both the dataset and source code made publicly available.

**Weaknesses:**

1. While the problem of numerical parameter constraints in the data science APIs discussed in this paper exists, it may not be particularly significant and appears to be easily resolvable. This raises concerns about the potential for further research.
2. Some of the experimental details of the paper are unclear and require additional elaboration.

**Questions:**

1. I believe that by incorporating API documentation into fine-tuning or including it within prompts, LLMs might better understand numerical parameter constraints, which could potentially address the issue you're exploring.

2. API parameters can vary across different versions of a library. Could these variations impact the results of your study?

3. Why do you choose to conduct the study with 28 APIs, whereas only 12 APIs are used in the DSEVAL dataset? What accounts for this difference?

4. The specific LLM used during input creation is not mentioned. Could you clarify which LLM is employed?

5. In Figure 5, why do different sets of APIs get affected as the input rank and dimensionality increase? Furthermore, how do rank and dimension affect the complexity of these APIs?

**Limitations:**

The authors have not dedicated a separate section to discussing the limitations; however, they consider the limitations to be associated with the scope of their study. I consider this sufficient.

---

> ### Author Rebuttal · Authors · 2024-08-07
>
> **Question-1: I believe that by incorporating API documentation into fine-tuning or including it within prompts, LLMs might better understand numerical parameter constraints, which could potentially address the issue you're exploring.**
>
> Thanks for this suggestion. We conducted additional experiments using the documentation-augmented setting across the 3 difficult API/constraints used in our CoT experiments. We provide the raw documentation of each API (obtained from the source code docstring) in the prompt and apply both base and instruction-following LLMs.
>
> In Figure 14 (see attached PDF in global response), we compare the performance with and without documentation. We found that there are cases where documentation can improve performance. For example, in the most difficult setting of `torch.nn.Conv2d`, adding documentation is able to improve performance of CodeLlama-34B-Instruct from 20% to 45% accuracy (Figure 14 (d)). However, there are also similar cases where adding documentation decreases performance. For example the GPT-4-Turbo-Instruct performance falls from 57.5% to 22.5% in the most difficult setting of `torch.nn.MaxPool2d` (Figure 14 (b)).
>
> Since we provide the raw documentation text without further processing, the success rate of adding documentation can vary depending on the specific model as well as the quality of the documentation. As such, this demonstrates that naively adding API documentation cannot always achieve better performance on our tasks.
>
> Again, we want to emphasize that we investigate the LLMs’ ability to implicitly reason about such constraints, given that LLMs have been trained on massive parameter combinations. Our findings in the paper shows that current LLMs cannot implicitly reason with more complex API constraints or unusual inputs. This can inspire many follow-up works on improving LLM performance, including the reviewer’s suggestion of incorporating API documentation or performing further fine-tuning.
>
> **Question-2: API parameters can vary across different versions of a library. Could these variations impact the results of your study?**
>
> Great question! Indeed, some APIs are not stable and their parameters and constraints could vary across versions. However, in this study we focus on the core functional APIs which should remain stable across library versions. We manually examined all 28 APIs across a 2 year period of major releases and found that only one API (`torch.squeeze`) received a small update to its parameter (removing the optional `out` parameter used to store the output tensor) while its numerical parameter constraints stayed the same. Therefore, we believe the API parameter variations across library versions have minimal impact on our study.
>
> **Question-3: Why do you choose to conduct the study with 28 APIs, whereas only 12 APIs are used in the DSEVAL dataset? What accounts for this difference?**
>
> Our 28 APIs are obtained by first selecting the 22 core APIs that have numeric parameter constraints in NNSmith, and adding additional 6 core APIs by examining the API list. Our chosen APIs are the core API commonly used by users. While there are a large number of APIs in DS libraries, the commonly used ones (e.g., Conv2d) are not that many. For example, the widely-used benchmark DS-1000 [29] contains 1000 DS problems collected from StackOverflow problems (including 68 PyTorch problems and 220 NumPy problems), reflecting realistic use cases; meanwhile, it only covers 16 PyTorch APIs and 59 NumPy APIs (after excluding data construction APIs like “np.ones”).
>
> Furthermore, some APIs have similar constraints or same constraint types. For example, `numpy.max` and `numpy.min` have the same constraints, whereas `torch.nn.Conv2d` and `torch.nn.Conv1d` have very similar constraints. Therefore, in our DSeval benchmark, we select 12 representative APIs to keep the experiments at an affordable scale, while still covering all major constraint categories in Table 1. We will clarify this further in the next revision of the paper.
>
> **Question-4: The specific LLM used during input creation is not mentioned. Could you clarify which LLM is employed?**
>
> Sorry for the confusion. We do not use LLMs during input creation. Instead, we have a template-based prompt and we generate random input values using built-in random number generators. We then use the Z3 constraint solver to filter out the invalid inputs and ensure that every input has at least one valid answer. For more details, please refer to Section 2.3.
>
> **Question-5: In Figure 5, why do different sets of APIs get affected as the input rank and dimensionality increase? Furthermore, how do rank and dimension affect the complexity of these APIs?**
>
> Input rank or dimensionality can affect different APIs depending on the type of numeric constraint of each API. Table 1 shows the categorization of the different types of constraints. For example, an API like `torch.nn.SoftMax` that has a constraint of $-rank \leq dim < rank$ will have its difficulty influenced by the actual rank of the input tensor. On the other hand, an API like `torch.nn.Conv2d` has a constraint of $in\underline{ }channels \% groups = 0$, which depends on the actual dimension value of the input (i.e., `in_channels`). As the dimension value of `in_channels` increases, it will be more difficult to select the `groups` parameter that can divide it evenly. Therefore, we increase the difficulty of different APIs based on whether the constraint depends on the rank, dimension, or both.
>
> Thanks for the question and we will clarify this further in the next revision of the paper.

---

> ### Comment · Area_Chair_CKy5 · 2024-08-12
>
> A quick reminder: Did the author's response adequately address your questions and concerns? Can you please write a brief response to let them know that you read it?

---

### Official Review · Reviewer_YxwD · 2024-07-05

**Soundness:** 2
**Presentation:** 3
**Contribution:** 1
**Rating:** 3
**Confidence:** 3

**Summary:**

This paper investigates a problem, precisely the title, can LLMs implicitly learn numeric parameter constraints in data science APIs? To investigate this problem, this paper constructs a benchmark, DSEVAL, which contains a series of API calling based code completion tasks. Then, this paper evaluates several LLMs to derive the conclusion: LLMs can implicitly learn the constraint pattern during training, but lack genuine comprehension of the underlying constraints.

**Strengths:**

- The investigated problem is interesting.
- The technique is sound.
- The writing is good.

**Weaknesses:**

- I am not sure whether this paper should be submitted to Dataset & Benchmark track, since this paper focuses on evaluating LLMs’ underlying capabilities for satisfying parameter constraints when calling data science APIs, rather than proposing any novel algorithmic advances, analysis, or applications.

- This paper should provide some naïve solutions as the starting point for this problem. However, this paper just provides a comprehensive evaluation on existing LLMs, without proposing any possible solution for this problem.

- I think one natural question is that whether LLMs can solve this problem by using ReAct [1] prompt strategy, i.e., each time LLMs call a data science API, LLMs are prompted to first output a thought about the constraint for this API, and then generate the parameter. I am curious about the results about this setting.

[1] ReAct: Synergizing reasoning and acting in language models, ICLR 2023.

**Questions:**

See weakness.

**Limitations:**

I donot find any discussion about limitation in this paper.

---

> ### Author Rebuttal · Authors · 2024-08-07
>
> **Question-1: I am not sure whether this paper should be submitted to Dataset & Benchmark track, since this paper focuses on evaluating LLMs’ underlying capabilities for satisfying parameter constraints when calling data science APIs, rather than proposing any novel algorithmic advances, analysis, or applications.**
>
> We thank the reviewer for this suggestion. We would like to point out that although there is a separate Dataset & Benchmark track, the main conference still accepts a primary area for evaluation: “Evaluation (methodology, meta studies, replicability and validity)”. Given the wide usage of DS libraries/APIs, automatically synthesizing valid DS programs with LLMs has been a critical research area in order to improve DS development efficiency [29] or improve the reliability of ML systems [16]. Therefore, it is critical to rigorously evaluate whether LLMs can implicitly learn the numeric constraints in data science APIs -- an untested assumption of LLMs’ capabilities. Our paper aims to propose a novel fine-grained analysis for DS API generation by isolating each API call and API parameter generation, as opposed to the previous coarse-grained benchmarks such as DS-1000. We believe our proposed evaluation methodology is novel and would be highly beneficial for the research community to accurately evaluate and analyze the LLMs’ ability in the important domain of DS code generation.
>
> Also, we do not aim to propose novel code generation algorithms in this work, but rather to point out interesting results and inspire future work. Some prior work [a,b,c,d] with similar contributions (i.e., discovering important findings/limitations for LLMs) were also published at NeurIPS main track.
>
> **Question-2: This paper should provide some naïve solutions as the starting point for this problem. However, this paper just provides a comprehensive evaluation on existing LLMs, without proposing any possible solution for this problem.**
>
> We understand that there are strategies at inference time such as ReAct which can boost the performance of LLMs. However, the goal of this paper is to investigate the ability of LLMs to ***implicitly*** model the parameter constraints using zero-shot inference, without adding ***explicit*** reasoning steps such as recalling the constraints with ReAct. Different models may perform differently depending on prompting techniques used, and the performance also highly depends on the wording and in-context demonstrations of the prompt. On the other hand, evaluating them under a naïve zero-shot auto-completion setting provides a fair assessment of the ability of the LLM to implicitly learn the numeric constraints.
>
> Meanwhile, please kindly note that we have applied Chain-of-Thought (CoT) prompting strategy to elicit thought steps at **the end of Section 4.3**. More specifically, we evaluate instruction-tuned LLMs using CoT prompting (“Please think step by step”) on multiple particularly difficult DS APIs. We found that although CoT prompting does help improve performance (especially when using state-of-the-art GPT-4-Turbo), it still struggles for more complex arithmetic constraints such as the ones in `torch.nn.Fold` (less than 5% accuracy in more difficult settings).
>
> **Question-3: I think one natural question is that whether LLMs can solve this problem by using ReAct [1] prompt strategy.**
>
> Thanks for the great suggestion! Although techniques that elicit explicit reasoning at inference time (like ReAct) is not the main focus of this paper, we agree that such an analysis is valuable and can provide additional insights to the goal of this paper.
>
> We applied the ReAct prompt strategy on the difficult APIs/constraints studied using CoT in the paper. Our ReAct prompt setup follows the reviewer’s suggestion (i.e., asking the LLM to generate a thought first then the code output). We also provide the LLM with a single demonstration of the ReAct task.
>
> Figure 12 (see attached PDF in global response) compares the results of the instruction-following LLMs using CoT versus ReAct as well as their base variants:
> - We see that for `torch.nn.MaxPool2d`, ReAct prompting generally performs better than CoT especially in more difficult problem settings (e.g., at highest difficulty setting, GPT-4-ReAct: 89.5% GPT-4-CoT: 56.0%). This demonstrates the effectiveness of ReAct in generating thoughts that can help with the correct API parameter generation.
> - However, for `torch.nn.Conv2d`, ReAct performs similarly to CoT prompting. The reason is that the constraint used in Conv2d is much more complex, requiring factorization. As such, smaller open-source LLMs cannot perform well even with reasoning steps. On the other hand, state-of-the-art LLMs like GPT-4-Turbo show their powerful reasoning abilities by improving the performance over the base variant with both CoT and ReAct.
> - Although ReAct performs better than CoT for the easier difficulty settings in `torch.nn.Fold`, its performance quickly drops in higher difficulty settings (at best ~5% accuracy with the best GPT-4-Turbo). Overall, this experiment results demonstrate that even more advanced prompting methods such as ReAct still cannot effectively handle more complex constraints.
>
> We thank the reviewer again for this suggestion and will work towards adding these results to the new revision of the paper.
>
>
> **References**
>
> [a] Testing the General Deductive Reasoning Capacity of Large Language Models Using OOD Examples. https://openreview.net/forum?id=MCVfX7HgPO NeurIPS 2023 (poster).
>
> [b] Are emergent abilities of large language models a mirage? https://openreview.net/forum?id=ITw9edRDlD NeurIPS 2023 (oral).
>
> [c] Statistical knowledge assessment for large language models. https://openreview.net/forum?id=pNtG6NAmx0 NeurIPS 2023 (poster).
>
> [d] Exploring length generalization in large language models. https://openreview.net/forum?id=zSkYVeX7bC4 NeurIPS 2022 (poster).

---

> > ### Comment · Reviewer_YxwD · 2024-08-10
> >
> > Thank the authors for providing the detailed feedback for my concerns.
> >
> > I have read the provided references but I think this paper fails in providing sufficient insights in evaluation, especially compared with [1, a-d].
> >
> > This paper investigates an interesting yet less significant problem, i.e., whether LLMs implicitly learn the numeric constraints in data science APIs. From my perspective, the answer is predictable: LLMs can implicitly learn simple constraints but fail for complex constraints during pre-training. This paper just verifies this point by (1) constructing a benchmark; and (2) testing the LLMs. However, I expect more interesting and significant insights such as **(a) the potential reasons for this phenomenon**; **(b) some possible initial solutions for this issue during pretraining / finetuning / inference stage**. I think exploring these directions may bring much more contribution to the community.
> >
> > Considering the points above, I do not think this paper provides enough contribution for a top conference like NeurIPS and I decide to keep my score unchanged.
> >
> > [1] Case-Based or Rule-Based: How Do Transformers Do the Math? ICML 2024.

---

> ### Author Response · Authors · 2024-08-11
>
> Thanks for reviewing our response and providing further feedback. We would like to politely point out that in our initial rebuttal response, we addressed all the reviewer’s initial questions and concerns. This included adding the requested experimental results demonstrating that even advanced prompting techniques like ReAct, cannot handle complex constraints. Please let us know which specific responses in the initial rebuttal that the reviewer still has issues with. Additionally, we also respectfully disagree with the reviewer’s new comments. Please see our detailed responses below:
>
> >This paper investigates an interesting yet less significant problem
>
> **We believe that evaluating the LLMs’ ability to generate correct and valid DS programs is extremely important**. Data science applications, and by extension DS libraries/APIs, are used daily by developers to process and analyze large amounts of data to build ML systems and make decisions. As such, DS APIs have penetrated almost every corner of modern society in the era of deep learning, including autonomous driving software [9, 27, 45], financial systems [18, 4], and coding assistants [44, 36]. LLMs are being widely used to aid in and generate programs, especially in the important domain of DS code. A recent study [e] has shown that GPT-4 can achieve performance on par with humans on various data analysis tasks. Additional work [f] has demonstrated improved performance when using LLMs to assist data analysts. Furthermore, there has been active research like [g] and the development of automatic tools like Data-Copilot [h] that use LLMs to automatically solve data science tasks. Due to this wide adoption, it is critical to check whether LLMs can implicitly learn the numeric constraints in data science APIs. While fine-grained analysis has been done in the math reasoning domain (e.g., Hu, Yi, et al.’s ICML 2024), none of the prior work has focused on the DS code generation domain. In this work, we advocate for a thorough examination of the assumption -- LLMs can implicitly learn the correct DS API parameter constraints, relied on but not tested by many prior works [46, 21, 16].
>
> Besides practical importance, we would also emphasize the notable distinctions in our approach and findings compared to previous work. **Our problems with DS API parameters are fundamentally different from the synthetic datasets or toy problems studied in prior work like Hu, Yi, et al.’s ICML 2024 work or [a,b,c,d], which we believe provides unique insights and values**. Firstly, our settings do not involve unnatural synthetic problems like linear regression function learning in [Hu, Yi, et al.] that are rarely seen during pre-training. Instead, there are more than 2,400,000/480,000 open-source projects using NumPy/PyTorch [i, j], meaning there exists a massive amount of DS API parameter examples in the pre-training dataset in an exact or very similar format as in our benchmark. Secondly, prior benchmarks typically present problems explicitly (e.g., `1+2=?`), while in our problem the constraints are implicit. This provides a unique testbed and poses an important theoretical question: Given a large set of examples $(\mathbf{X_1},\mathbf{X_2},\mathbf{X_3}, \ldots, \mathbf{X_N})$ where each assignment $\mathbf{X_i}=(x_i^1,x_i^2,...x_i^m)$ satisfies a certain constraint $\phi$ in first-order logic, can a model ***pre-trained*** on large mixture of data including these examples (i.e., $\\{\mathbf{X_i}\\}_{i=1}^N$) implicitly learn $\phi$ and generalize to predict $x_i^j$ when $x_i^{-j}$ is out of distribution?
>
> **References**
>
> [e] Is GPT-4 a Good Data Analyst? https://arxiv.org/abs/2305.15038 (2023).
>
> [f] How Do Data Analysts Respond to AI Assistance? A Wizard-of-Oz
> Study? https://arxiv.org/abs/2309.10108 (CHI 2024).
>
> [g] Large Language Models for Automated Data Science: Introducing CAAFE for Context-Aware Automated Feature Engineering. https://arxiv.org/abs/2305.03403 (NeurIPS 2023).
>
> [h] Data-Copilot: Bridging Billions of Data and Humans with Autonomous Workflow. https://arxiv.org/abs/2306.07209 (2023).
>
> [i] https://github.com/pytorch/pytorch/network/dependents
>
> [j] https://github.com/numpy/numpy/network/dependents

---

> ### Author Response · Authors · 2024-08-11
>
> > From my perspective, the answer is predictable:  LLMs can implicitly learn simple constraints but fail for complex constraints during pre-training
>
> **Despite the seemingly obvious answer, our result reveals multiple surprising findings**. For example, Figure 13 (see attached PDF in global response) shows that for the simple equality constraint: `in_features=input.shape[-1]` in `torch.nn.Linear`, open-source LLMs cannot generalize to out-of-distribution problems (i.e., tensor rank > 4) and have less than 20% accuracy. In contrast, GPT-4-Turbo is able to maintain its performance. One potential reason is that LLMs are mostly trained with data with very common shapes or ranks (e.g., rank=3, 4), leading them to always copy the 3rd/4th shape dimension, while the true rule is to copy the last dimension. As such, LLMs can easily make mistakes on uncommon inputs. For example, when rank=6, they still copy the 3rd or 4th dimension. We also examined the attention weights and found that when LLMs make mistakes for `torch.nn.Linear`, it is usually because they are assigning larger attention weights to incorrect positions, and therefore copying values from a wrong parameter. We believe there can be lots of interesting follow-up works from our fine-grained benchmark and analysis.
>
> > (a) the potential reasons for this phenomenon
>
> Please also note that we did conduct in-depth analyses based on different problem settings, constraint types (e.g., equality, inequality, arithmetic, and set-related), and across LLMs, and have interesting findings (highlighted in Section 4.1, 4.2, and 4.3). For example, we found that there is a large difference between open source LLMs and state-of-the-art LLM’s performance, especially on complex constraints. Such difference in performance has not been observed previously on standard coding benchmarks (e.g., HumanEval) where the performance gap between open and close source LLMs are considerably smaller. We also have additional interesting findings and analyzed potential reasons regarding the common mistakes made by LLMs, such as:
>
> 1. **LLMs struggle with uncommon input tensors**: We found that across many APIs and constraints, LLMs struggle when provided with uncommon input tensor ranks (i.e., rank > 4) or uncommon shapes (e.g., `x = torch.rand(9, 30, 23, 4)`). The reason is that LLMs are mostly trained with data that contains very common shapes or ranks. As such, LLMs can easily make mistakes on uncommon inputs.
>
> 2. **LLMs tend to predict common parameter values blindly**:  We also observe that LLMs tend to generate common parameter values (e.g., 0, 1, powers of 2) which often turn out to be incorrect. This is again because LLMs are trained with pre-training code that frequently contains such parameter patterns and thus are likely to predict them even given a different input context.
>
> 3. **LLMs pay attention to the wrong tokens/irrelevant parameters**: LLMs can learn spurious correlations and pay attention to the wrong context tokens. For example, open-source LLMs struggle with the simple equality constraint `in_features=input.shape[-1]` in `torch.nn.Linear` because the attention weights are focused on the irrelevant parameters.
>
> > (b) some possible initial solutions for this issue during pretraining / finetuning / inference stage
>
> Please note that we have explored possible initial solutions such as prompting techniques like CoT during the inference stage. We also added the new ReAct and documentation-augmented experiments as suggested by reviewers.
>
> We also want to point out that the exact reason for investigating this problem is because the model has been exposed to a large amount of similar data to the problems in our studied settings during the pre-training stage. This led to our goal of studying the ***implicit*** ability for LLMs to satisfy numerical constraints across different constraint types and difficulty levels without any additional fine-tuning or prompting.
>
> Additionally, we believe that before exploring additional new techniques to improve performance, building an extensive and high quality benchmark is extremely important. To enable future research and evaluate potential solutions, we constructed a detailed benchmark -- DSeval. We believe this is an essential and crucial contribution, allowing new techniques to be developed and tested in this important domain of generating correct and valid DS code.

---

> > ### Comment · Reviewer_YxwD · 2024-08-12
> >
> > Thanks for your further clarification. However, I still have the following main concerns.
> >
> > **1. The mentioned surprising findings do not seem that surprising for me, which is also the main reason why I think that this paper investigates an interesting yet less significant problem – the key finding in this paper (i.e., LLMs can implicitly learn simple constraints but fail for complex constraints during pre-training) is predictable.**
> >
> > > (1) LLMs struggle with uncommon input tensor. (2) LLMs tend to predict common parameter values blindly.
> >
> > These two findings can be indicated by [1]. LLMs perform case-based reasoning instead of rule-based reasoning: LLMs struggle with uncommon input tensor due to lack of similar case, and LLMs tend to generate common parameter values due to similar case in pretraining.
> >
> > [1] Case-Based or Rule-Based: How Do Transformers Do the Math? ICML 2024.
> >
> > > (3) LLMs pay attention to the wrong tokens/irrelevant parameters.
> >
> > Could you please tell me which part of your paper can lead to this finding? I do not find any attention map visualization in this paper. Please correct me if I am wrong.
> >
> > **2. The assumed potential reasons for these findings should be supported by experiments.**
> >
> > > LLMs struggle with uncommon input tensors: We found that across many APIs and constraints, LLMs struggle when provided with uncommon input tensor ranks (i.e., rank > 4) or uncommon shapes (e.g., x = torch.rand(9, 30, 23, 4)). The reason is that LLMs are mostly trained with data that contains very common shapes or ranks. As such, LLMs can easily make mistakes on uncommon inputs.
> >
> > For example, I think providing the experiments of pretraining/finetuning LLMs with the designed uncommon inputs can support the assumed reasons.
> >
> > **3. The original manuscript only provides CoT as the initial solution, while providing ReAct and documentation-augmentation during the rebuttal. All these three methods are performed during inference stage. How about pretraining/finetuning stage?**
> >
> > - If Figure 6h refers to the CoT results, I think that the legends are wrong? Or could you tell me which line refers to CoT?
> >
> > - I think documentation augmentation can be a strong strategy. But from Figure 14, GPT-4 turbo w/ doc performs poor compared with GPT-4 turbo. This is quite surprising for me. Could you please provide the prompt for GPT-4 turbo w/ doc? Moreover, could you tell me the difference between gpt4-turbo and gpt4-turbo-inst.? I think that gpt4-turbo from OpenAI is already the instruction-following version.

---

> ### Author Response · Authors · 2024-08-12
>
> We truly appreciate the prompt response! Meanwhile, we noticed some misunderstandings and additional experiment requests which were not mentioned in the original review (so we did not have a chance to address in the rebuttal period). We are happy to respond to them in detail below.
>
> >The mentioned surprising findings do not seem that surprising for me, which is also the main reason why I think that this paper investigates an interesting yet less significant problem – the key finding in this paper (i.e., LLMs can implicitly learn simple constraints but fail for complex constraints during pre-training) is predictable. ... These two findings can be indicated by [1].
>
> Please note that in our previous response, we referred to [1] as Hu, Yi, et al.’s ICML 2024 or [Hu, Yi, et al.] and have already discussed the fundamental difference. We will make sure to discuss this related work in our revised manuscript.
>
> Again, we would like to point out that our work and [1] study two completely different problem domains:
>
> 1. First, different from the **explicit** problems in [1] (e.g., `1+2=?`), in our study, the API constraints are **not** directly specified in the problem. Instead the LLM should **implicitly** learn the constraints through pre-training on large amounts of open-source DS code examples. This makes the fundamental problem setup different between our work and [1].
>
> 2. Second, our problems are also completely different from the synthetic problems (e.g., chicken and rabbit problem) used in [1] that are rarely seen during pre-training. Instead, there are more than 2,400,000/480,000 open-source projects using NumPy/PyTorch, meaning there exists a massive amount of DS program examples in the pre-training dataset in an exact or very similar format as in our benchmark. This makes our study’s starting point completely different from prior work [1].
>
> 3. Third, we would like to point out that there are other works similar to [1] which show completely different findings. For example [1] shows that scratchpad fine-tuning underperforms compared to direct fine-tuning, while another work [a] demonstrates that fine-tuning with scratchpad (especially with few-shot examples) can significantly improve length generalization in the coding domain.
>
> Furthermore, we kindly argue that the value of scientific progress and discovery is not measured by the predictability of the final conclusion. In our work, we perform a rigorous study on evaluating the implicit ability of LLMs to satisfy valid numeric constraints in DS programs, which is an extremely important problem in the era of deep learning. We believe that even if some of the main conclusions are predictable, we are the first one to demonstrate this for the important domain of DS code generation. Additionally, our detailed analysis of each LLM’s performance on different problem settings and difficulties, along with our insights, can provide concrete guidelines for improving code LLMs and inspire future work.
>
> **References**
>
> [a] Exploring Length Generalization in Large Language Models. https://openreview.net/forum?id=zSkYVeX7bC4 NeurIPS 2022 (poster).

---

> ### Author Response · Authors · 2024-08-12
>
> > (3) LLMs pay attention to the wrong tokens/irrelevant parameters. Could you please tell me which part of your paper can lead to this finding? I do not find any attention map visualization in this paper. Please correct me if I am wrong.
>
> We apologize for the confusion. The attention analysis is part of the additional in-depth error analysis we conducted during the rebuttal period following the suggestion of Reviewer xaud.
>
> Since we couldn’t provide figures in the comments, we explain the results in text below. We will add the attention visualization as well as more in-depth discussion in our revision.
>
> To give an example, for the DeepSeek-1.3B model and the API `torch.nn.Linear` with difficulty of rank=5, the input tensor has 5 dimensions and is presented in context like `x = torch.randn(5, 4, 2, 14, 11)`, and the model is asked to predict a parameter `in_features`, with the constraint being `in_features==input.shape[-1]`. For this simple task, the LLM only achieves 6% accuracy. To investigate why, we compute the attention weight of each input token (maximum of all attention heads across all layers) to locate the most significant token among the relevant ones (e.g., `5, 4, 2, 14, 11` in the previous example). Next, we map it to a specific dimension of the input tensor (i.e., `0, 1, 2, 3, 4`). Here’s the result:
>
> Out of the 200 tests, here is the number of times that the predicted value matches each input dimension: `{0: 30, 1: 113, 2: 40, 3: 19, 4: 12}`. The correct answer should be `4`, the last dimension of a rank-5 tensor. Note that the sum of this is not 200 because the predicted value can match more than one input dimensions if they are identical.
>
> We observe that the predicted value almost always (98%) matches with the context token that has the highest attention weight. This indicates that the LLM does learn to always copy a specific dimension from the input tensor. As for the detailed break-down: (1) When the attention is paid to the wrong context token/parameter value (92%), it leads to incorrect results; (2) For 1% of the times, it pays attention to the right dimension; (3) Interestingly, for 5% it copies from the wrong position but that specific value just happens to be correct. Please kindly let us know if any further analysis is needed, and we are happy to add that in the next revision.
>
>
> >The original manuscript only provides CoT as the initial solution, while providing ReAct and documentation-augmentation during the rebuttal. All these three methods are performed during inference stage. How about pretraining/finetuning stage? For example, I think providing the experiments of pretraining/finetuning LLMs with the designed uncommon inputs can support the assumed reasons.
>
> Again, we want to stress that our goal is to evaluate the ***implicit*** reasoning capability of LLMs in solving numeric constraints (as indicated by our title as well as introduction paragraphs). We totally agree with the reviewer that having training experiments can potentially reinforce and unlock additional insights. However, this is ***outside the scope of our study***. Additionally,  the reviewer only asked for the additional ReAct prompting experiment for the rebuttal period, which we did. Due to the time limit and resource cost of the reviewer-author-discussion period we are unable to perform any pre-training or fine-tuning experiments.
>
> Furthermore, prior work [b, c, d] with similar contributions (i.e., discovering important findings/limitations for LLMs) also do not include any pre-training or even fine-tuning experiments. We would also like to point out that fine-tuning and pre-training with task-specific synthetic data on small-sized LLMs do not always lead to the same conclusion when done on large state-of-the-art LLMs trained with a mixture of different real-world data.
>
>
> **References**
>
> [b] Statistical Knowledge Assessment for Large Language Models. https://openreview.net/forum?id=pNtG6NAmx0 NeurIPS 2023 (poster).
>
> [c] Towards Understanding Factual Knowledge of Large Language Models. https://openreview.net/forum?id=9OevMUdods ICLR 2024 (poster).
>
> [d] KoLA: Carefully Benchmarking World Knowledge of Large Language Models. https://openreview.net/forum?id=AqN23oqraW ICLR 2024 (poster).

---

> ### Author Response · Authors · 2024-08-12
>
> >If Figure 6h refers to the CoT results, I think that the legends are wrong? Or could you tell me which line refers to CoT?
>
> The dashed lines with triangular markers in Figure 6h refer to the CoT prompting approach when using the instruction-tuned LLMs (as described in line 296-297). We included the complete CoT prompt used by us for all models in Appendix. The solid lines with circular markers refer to the base LLMs (not the instruction-tuned variants), allowing us to compare and contrast their performances.
>
> >I think documentation augmentation can be a strong strategy. But from Figure 14, GPT-4 turbo w/ doc performs poorly compared with GPT-4 turbo. This is quite surprising for me. Could you please provide the prompt for GPT-4 turbo w/ doc?
>
> Great observation! We would like to point out that adding documentation does not always decrease performance. For example, in the `torch.nn.Fold`, adding documentation is able to improve performance of GPT-4-Turbo across all difficulty levels (Figure 14 (e)). However, like the reviewer pointed out, there are also similar cases where adding documentation decreases performance. For example the GPT-4-Turbo drops in performance when given the documentation for `torch.nn.Conv2d` (Figure 14 (c)).
>
>
>
> Please note that the success rate of adding documentation can vary depending on the specific model, API studied, as well as the quality of the documentation. For example, although modern LLMs are more and more powerful, they are still far from perfect and cannot always effectively leverage all the information provided. Such “surprising” findings can hopefully also inspire various future work for further improving the ***implicit*** reasoning capability of LLMs.
>
> Here is the complete prompt for GPT-4-Turbo for `torch.nn.Fold` with documentation:
>
> System Prompt:
> ```
> You are an expert Python programmer and are good at writing correct PyTorch code.
> ```
>
> Input Prompt:
> ````
> Please refer to the given API documentation and complete the Python program.
> Documentation for the torch.nn.Fold API:
> {documentation_omitted_due_to_space}
>
> Please complete the program by filling in the correct API parameter(s). You should keep the exact same program unchanged, just fill in the missing code part.
> ```python
> import torch
> x = torch.randn(90, 96, 126)
> m = torch.nn.Fold(output_size=(15, 10), kernel_size=<insert code here>, dilation=1, padding=0, stride=1)
> ```
> ````
>
> In the above example, we omit the exact document due to the reply word limit. We provide the raw documentation of each API (obtained from the source code docstring). An example can be found here: https://pytorch.org/docs/stable/_modules/torch/nn/modules/fold.html#Fold
>
> >Moreover, could you tell me the difference between gpt4-turbo and gpt4-turbo-inst.? I think that gpt4-turbo from OpenAI is already the instruction-following version.
>
> Sorry for the confusion, you are right that GPT-4-Turbo is already the instruction-following version. We use *-Inst. to differentiate the generation setting: infilling (GPT-4-Turbo) and free-form generation (GPT-4-Turbo-Inst.) used in our study. For GPT-4-Turbo, we use our infill-specific prompt (See Appendix F for an example) to ask it to **only fill in** the missing code without adding any additional text. This setup allows us to compare against other infilling LLMs in the same setting. On the other hand, for GPT-4-Turbo-Inst., we allow it to generate additional text (such as CoT or ReAct reasoning steps). Please note that GPT-4-Turbo-Inst. is only used in our CoT experiments in the original paper. Thanks again for the question, and we will definitely have better naming for the next revision to avoid such confusion.

---

> > ### Comment · Reviewer_YxwD · 2024-08-13
> >
> > Thanks for your further reply. I want to clarify some key points.
> >
> > **1. I do not think the raised concerns are clear misunderstandings.**
> >
> > I mention the reference [1] to demonstrate why I think this paper does not discover exciting findings for me. I didn’t claim that this paper and [1] investigate similar problems.
> >
> > [1] Case-Based or Rule-Based: How Do Transformers Do the Math? ICML 2024.
> >
> > **2. I didn’t request additional experimental results during author-reviewer discussion. Also, the concern about initial solutions is already included in my initial review (See Weakness 2).**
> >
> > The mentioned experiments are the reasons why I think this paper does not provide sufficient insights about potential reasons for the phenomenon.
> >
> > Although I acknowledge that the authors have made efforts during the rebuttal period, **from my perspective**, this paper does not meet the high standards expected at NeurIPS and my concerns about the contribution remain unresolved, so I will maintain my original score.

---

> > > ### Author Response · Authors · 2024-08-13
> > >
> > > We are very sad to hear that the reviewer believed our effort spent during multiple responses and added experimental results were not sufficient. We have made every effort possible to address all the main concerns raised, and hope the reviewer can also look at the other positive reviews and reconsider. Below are our answers:
> > >
> > > >I do not think the raised concerns are clear misunderstandings. I mention the reference [1] to demonstrate why I think this paper does not discover exciting findings for me. I didn’t claim that this paper and [1] investigate similar problems.
> > >
> > > If the reviewer believes that [1] and our work do not investigate similar problems then how does the finding of [1] affect the contribution of our work? Just like our previous replies, we again point out that scientific discovery is not based on whether something is “exciting” or can be easily “predicted”. The prediction (i.e., hypothesis) to be investigated needs to be rigorously shown and proven through experiments. We believe that our work is an extensive evaluation of the implicit ability of LLMs to satisfy valid numeric constraints in DS programs, which is an extremely important problem in the era of deep learning.
> > >
> > > > I didn’t request additional experimental results during author-reviewer discussion. Also, the concern about initial solutions is already included in my initial review (See Weakness 2).
> > >
> > > You did. The original review did not mention anything about fine-tuning or pre-training, even including Weakness 2. In fact the original review only requested a new experiment on ReAct in Weakness 3: *“I think one natural question is that whether LLMs can solve this problem by using ReAct [1] prompt strategy, i.e., each time LLMs call a data science API, LLMs are prompted to first output a thought about the constraint for this API, and then generate the parameter. I am curious about the results about this setting.”*. We did add this ReAct experiment and demonstrate interesting results, but then the reviewer completely ignored our new ReAct results, and asked for new experiments on fine-tuning/pre-training, e.g., in the new [comment on Aug 11](https://openreview.net/forum?id=LfC5rujSTk&noteId=rrQOuQhfQa): *“The original manuscript only provides CoT as the initial solution, while providing ReAct and documentation-augmentation during the rebuttal. All these three methods are performed during inference stage. How about pretraining/finetuning stage?”*.

---

### Official Review · Reviewer_bNWw · 2024-07-05

**Soundness:** 3
**Presentation:** 3
**Contribution:** 3
**Rating:** 6
**Confidence:** 4

**Summary:**

The authors systematically investigate how well current LLMs learn numeric
parameter constraints of functions and deep learning operators in the Numpy and
PyTorch libraries. Their main finding is that although it is widely assumed that
current LLMs can solve arithmetic constraints, the performance of even
state-of-the-art LLMs like GPT-4-Turbo drops drastically when the complexity of
the constraints increases, and accuracy sometimes even drops to 0.

Another interesting finding is that there is a huge gap between open source
models and GPT-4-Turbo, which is not captured by benchmarks like HumanEval. The
authors introduce a public benchmark called DSeval based on these findings which
demonstrates this gap and allows future research to measure and narrow it.

**Strengths:**

The paper presents a first, thorough and systematic study of these constraints,
which shows that in many cases LLMs perform worse than what is currently widely
believed. The paper also illuminates the strong and weak points of these LLMs,
and introduces a new benchmark dataset which shows a huge gap between
open-source LLMs and GPT-4-Turbo, even when they seem to perform similarly on
other benchmarks like HumanEval.

**Weaknesses:**

Some of the terminology is not aligned with the widely used meaning of the
concepts, which makes the paper harder to read. Most significantly, the paper
talks about APIs like sum, max, or reshape. These are usually considered
functions and/or deep learning operators , and not APIs. APIs are usually
considered to be larger, like the API of a library such as PyTorch, or the API
of OpenAI. Also, PyTorch is usually thought of as a deep learning library, not a
data science library.



Based on this I would probably not call the benchmark DSeval. Also, the
parameter constraints are probably also a salient feature of the benchmark and
the work, so I would call it something along the lines of DLParamEval, or
DLParamConstraintEval. Naming the benchmark is of course the authors's choice,
so this is just a suggestion and won't influence my score.

The paper doesn't distinguish between GPT-4 and GPT-4-Turbo. The authors usually
write about GPT-4 while GPT-4-Turbo is used in the evaluations.

Some smaller remarks:
- I believe that the last three lines of the input are actually part of the
  output in Figure 2a.
- The explanation of subfigures could be added to the caption of Figure 2
- full program, all parameters, and individual parameters could be explained
  briefly also near line 66

**Questions:**

In Figure 5, why does the accuracy drop much more significantly for Linear than
for the other functions/constructors? Is this only true for DeepSeekCoder-33B?

**Limitations:**

There is no section on limitations but I don't believe that one would be
necessary as the whole work is about the limitations of LLMs.

---

> ### Author Rebuttal · Authors · 2024-08-07
>
> **Question-1: In Figure 5, why does the accuracy drop much more significantly for Linear than for the other functions/constructors? Is this only true for DeepSeekCoder-33B?**
>
> Thanks for bringing it up! This is an interesting result. For `torch.nn.Linear(in_features, out_features)`, the only constraint is that the `in_features` should match the last dimension of the input tensor. However, the DeepSeekCoder-33B model tends to copy the wrong dimension of the input tensor, likely because it hasn't seen lots of high-rank tensors (rank > 4) in the pre-training data.
>
> We further evaluate this phenomenon across different LLMs. Figure 13 (see attached PDF in global response) shows the results for both the full API parameter and single API parameter setting for `torch.nn.Linear` as we increase the difficulty (rank of the input data) across 8 LLMs. Similar to DeepSeekCoder-33B, the performance of other LLMs also drops significantly when rank reaches 4. Afterwards, the performance stabilizes for higher difficulties (i.e., rank > 4) especially for open-source LLMs. This is true for both the full API parameter and single API parameter setting. Surprisingly, we found that even the state-of-the-art GPT-4-Turbo drops in performance when the rank reaches 4. However, we see that GPT-4-Turbo was able to improve its performance in higher difficulties (i.e., rank > 6). After looking at the results, we found that for lower ranks, GPT-4-Turbo tends to use other APIs as “short-cuts” and forgo the analysis on `torch.nn.Linear` directly as shown in an example below:
>
> ```python
> import torch
> x = torch.randn(1, 8, 10, 10)
> m = torch.nn.Linear(8*10*10, 3)
> y = m(x.view(1, -1))
> ```
>
> In the above example, instead of using only the last dimension (10), GPT-4-Turbo multiplies all the previous dimensions together and performs a flattening operation (`x.view(1, -1)`). This does not reflect the original meaning of the code and as such is evaluated as incorrect.
>
> Thanks for this great question again! We will add this experiment results in the next revision of the paper.
>
> **Question-2: Most significantly, the paper talks about APIs like sum, max, or reshape. These are usually considered functions and/or deep learning operators , and not APIs. APIs are usually considered to be larger, like the API of a library such as PyTorch, or the API of OpenAI. Also, PyTorch is usually thought of as a deep learning library, not a data science library.**
>
> Thanks for the suggestions! First, please note that deep learning/machine learning are generally considered as subfields of data science [48]. Libraries like PyTorch contain many data process, manipulation, and transformation APIs used for data science operations. Furthermore, the widely-used data science benchmark -- DS-1000 [29], contains both TensorFlow and PyTorch problems. Additionally, please note that we are not only targeting PyTorch, we also include NumPy in our study and we plan to generalize this study to more data science libraries in the future. For libraries like PyTorch and NumPy, indeed “smaller” APIs like `sum`, `max`, or `reshape` are called operators as well, and there are “larger” APIs like `SoftMax`, `BatchNorm`, and `Conv2d` which combine multiple operators; Meanwhile, they can be all considered APIs as well (please see https://pytorch.org/docs/stable/index.html - Python API). We use the term “API” instead of “operator”, because it is more general and accurately describes our target: the *publicly* exposed operators or functions for which we expect LLMs to generate correct parameters.

---

> > ### Comment · Reviewer_bNWw · 2024-08-12
> >
> > Thank you for your answers!
> >
> > It is indeed an interesting result, thank you for obtaining it!
> >
> > Regarding your second answer, I don't believe that Data Science is a subfield of
> > Machine Learning but that's probably a matter of opinion so it won't influence
> > my score.
> >
> > The PyTorch documentation also suggests that APIs are not single functions. If
> > you open it using the link you provided, you can immediately see that the
> > "Python API" refers to the collection of all of the functions, classes, etc.;
> > it's not the case that each function is an API.
> >
> > Similarly, if you go to the "Functional higher level API", for example, you can
> > see that the first sentence reads: "This section contains the higher level API
> > for the autograd that builds on the basic API above and allows you to compute
> > jacobians, hessians, etc.". So the API is the collection of the functions on
> > this page.

---

> > > ### Author Response · Authors · 2024-08-12
> > >
> > > Thanks for reading our response and sharing your thoughts!
> > >
> > > >The PyTorch documentation also suggests that APIs are not single functions. If you open it using the link you provided, you can immediately see that the "Python API" refers to the collection of all of the functions, classes, etc.; it's not the case that each function is an API.
> > >
> > > We acknowledge that there is a difference of opinion regarding the definition of an API in terms of DS libraries including PyTorch. Here we refer the reviewer to prior work [a, b, 32] that also considers each individual function/class/etc. as an API. We will definitely make our definition clear in the next revision of our paper.
> > >
> > > **References**
> > >
> > > [a] DocTer: Documentation-Guided Fuzzing for Testing Deep
> > > Learning API Functions. (ISSTA 2023)
> > >
> > > [b] Free Lunch for Testing: Fuzzing Deep-Learning Libraries from Open Source. (ICSE 2022)

---

> > > > ### Author Response · Authors · 2024-08-13
> > > > **Thanks for your review and great suggestions**
> > > >
> > > > Thanks again for your great suggestions (especially pointing out the interesting results on `torch.nn.Linear`) that helps us strengthen the paper! We would like to share our new interesting findings regarding the common mistakes made by LLMs. In additional to 1) struggling with uncommon input tensors; and 2) predicting common parameter values blindly, we found that 3) **LLMs pay attention to the wrong tokens/irrelevant parameters**: LLMs can learn spurious correlations and pay attention to the wrong context tokens. For example, open-source LLMs struggle with the simple equality constraint `in_features=input.shape[-1]` in `torch.nn.Linear` because the attention weights are focused on the irrelevant parameters.
> > > >
> > > > **Attention analysis:** To give an example, for the DeepSeek-1.3B model and the API `torch.nn.Linear` with difficulty of rank=5, the input tensor has 5 dimensions and is presented in context like `x = torch.randn(5, 4, 2, 14, 11)`, and the model is asked to predict a parameter `in_features`, with the constraint being `in_features==input.shape[-1]`. For this simple task, the LLM only achieves 6% accuracy. To investigate why, we compute the attention weight of each input token (maximum of all attention heads across all layers) to locate the most significant token among the relevant ones (e.g., `5, 4, 2, 14, 11` in the previous example). Next, we map it to a specific dimension of the input tensor (i.e., `0, 1, 2, 3, 4`). Here’s the result:
> > > >
> > > > Out of the 200 tests, here is the number of times that the predicted value matches each input dimension: `{0: 30, 1: 113, 2: 40, 3: 19, 4: 12}`. The correct answer should be `4`, the last dimension of a rank-5 tensor. Note that the sum of this is not 200 because the predicted value can match more than one input dimensions if they are identical.
> > > >
> > > > We observe that the predicted value almost always (98%) matches with the context token that has the highest attention weight. This indicates that the LLM does learn to always copy a specific dimension from the input tensor. As for the detailed break-down: (1) When the attention is paid to the wrong context token/parameter value (92%), it leads to incorrect results; (2) For 1% of the times, it pays attention to the right dimension; (3) Interestingly, for 5% it copies from the wrong position but that specific value just happens to be correct.
> > > >
> > > > We will incorporate your suggested `torch.nn.Linear` analysis, the attention visualization results, and other remarks in your review in our next revision. Thanks again for your support and valuable comments!

---

### Official Review · Reviewer_xaud · 2024-07-12

**Soundness:** 3
**Presentation:** 3
**Contribution:** 3
**Rating:** 6
**Confidence:** 3

**Summary:**

This paper investigates the ability of large language models (LLMs) to implicitly learn and apply numeric parameter constraints in data science (DS) APIs, focusing on PyTorch and NumPy libraries. The authors conduct a comprehensive study across 28 representative APIs, evaluating LLMs in three settings: full program generation, all parameter prediction, and individual parameter prediction. They introduce DSEVAL, a benchmark containing 19,600 problems across 12 APIs with varying difficulty levels. The study evaluates both open-source and closed-source LLMs, including state-of-the-art models like GPT-4. Results show that while LLMs perform well on simple constraints and common patterns, their performance degrades significantly with increased difficulty or unusual inputs. The authors conclude that current LLMs, including GPT-4, struggle with complex arithmetic constraints and often rely on memorization of common patterns rather than true understanding of the underlying constraints. This research highlights the limitations of LLMs in handling numeric API constraints and provides a benchmark for future improvements in this area.

**Strengths:**

Originality: First systematic study of LLMs' ability to handle numeric constraints in DS APIs. Propose a novel benchmark (DSEVAL) for evaluating this specific capability. Challenges previously untested assumptions about LLM capabilities

Quality: Comprehensive evaluation across 28 diverse APIs. Rigorous methodology using SMT solvers for validation. Evaluation of both open-source and closed-source models, including state-of-the-art LLMs

Clarity:. Well-structured presentation with clear explanations of complex concepts. Effective use of examples, tables, and figures to illustrate key points. Accessible to readers without deep expertise in LLMs or DS APIs

**Weaknesses:**

Absence of error bars or statistical significance tests: Lacks quantification of result certainty. Suggestion: Include error bars or conduct statistical tests to strengthen the validity of findings

Limited exploration of prompting techniques: Minimal investigation of advanced prompting methods. Suggestion: Experiment with more diverse prompting strategies to potentially improve LLM performance.

Lack of human baseline: No comparison to human performance on similar tasks. Suggestion: Include a human baseline to contextualize LLM performance

Limited discussion on potential solutions: Doesn't offer many concrete suggestions for improving LLMs in this domain. Suggestion: Propose and discuss potential approaches to enhance LLM capabilities in handling numeric constraints

**Questions:**

1. Why did you limit your study to PyTorch and NumPy? Wouldn't including more libraries make your findings more generalizable?

2. Your study lacks a human baseline. How can we interpret the LLM performance without knowing how humans perform on similar tasks?

3. You don't provide error bars or statistical significance tests. How confident are you in the reliability of your results?

4. Your paper doesn't explore fine-tuning or more advanced prompting techniques. Couldn't these potentially improve LLM performance significantly?

5. The paper lacks an in-depth error analysis. Wouldn't categorizing common mistakes provide more insights into LLM limitations?

**Limitations:**

Lack of human baseline: The study does not provide a comparison to human performance on similar tasks, making it challenging to contextualize the LLM performance.

Absence of error analysis: The study lacks a detailed categorization and analysis of common error patterns, which could provide deeper insights into specific LLM limitations.

No error bars or statistical significance tests: The paper does not include measures of statistical uncertainty, which could strengthen the validity of the findings.

Limited discussion on potential solutions: While the paper identifies limitations in LLM performance, it offers few concrete suggestions for improving LLM capabilities in handling numeric constraints.

---

> ### Author Rebuttal · Authors · 2024-08-07
>
> **Question-1: Why did you limit your study to PyTorch and NumPy?**
>
> We chose PyTorch and NumPy due to their wide adoption in the data science community: PyTorch is used by over 480k open-source GitHub projects [a] and NumPy is installed more than 300 million times monthly [b]. Focusing on these two allowed for a more in-depth analysis within our budget. You are absolutely right that including more libraries will make our findings more generalizable, and we plan to include more DS libraries (e.g., Pandas, Matplotlib, Scikit-learn, SciPy, and Tensorflow) in future work.
>
> **Question-2. Your study lacks a human baseline. How can we interpret the LLM performance without knowing how humans perform on similar tasks?**
>
> Please note that most code benchmarking work (e.g., DS-1000, MBPP, HumanEval) does not provide a human baseline. This is because it is expensive to conduct user studies on coding problems, especially for DS programs where domain expertise is required. Additionally, people with different experiences can have vastly different performances. Meanwhile, unlike prior benchmarks where the natural language problem description can be unclear and thus imposes an upper bound on LLM performance [c], our benchmark is unambiguous, and we could expect ~100% accuracy from sufficiently strong LLMs or expert developers.
>
> In this work, we interpret the LLM performance by comparing different LLMs and different problem complexities. As nicely summarized by Reviewer-bNWw:
>
> > (1) Their main finding is that although it is widely assumed that current LLMs can solve arithmetic constraints, the performance of even state-of-the-art LLMs like GPT-4-Turbo drops drastically when the complexity of the constraints increases, and accuracy sometimes even drops to 0.
>
> > (2) Another interesting finding is that there is a huge gap between open source models and GPT-4-Turbo, which is not captured by benchmarks like HumanEval.
>
> **Question-3: You don't provide error bars or statistical significance tests. How confident are you in the reliability of your results?**
>
> Due to the expensive costs, we use greedy decoding in all our experiments except for the full program setting (Section 4.1). Therefore, in all other settings, the performance is deterministic.
>
> For the full program setting, we vary the temperature from 0.2 to 1 and sample 200 DS programs across all 28 APIs. As a result, performing repeated experiments to compute error bars will be extremely costly especially for commercial models like GPT-4-Turbo.
>
> Furthermore, please note that in all settings of our benchmark, each problem is independently sampled from an identical distribution of its difficulty level. As such, the pass/fail outcome follows a Bernoulli distribution, and we draw N=200 samples for each task distribution. Therefore, using the normal approximation: $p\approx
> \hat{p}\pm\frac{z_{\alpha}}{\sqrt{N}}\sqrt{\hat{p}(1-\hat{p})}$, where $\hat{p}$ is the average accuracy (proportion of successes), $z_{\alpha}$ is the $1-\frac{\alpha}{2}$ quantile of a standard normal distribution corresponding to the target error rate $\alpha$. For a 95% confidence level, $z_{.05}=1.96$ and $p$ (i.e., accuracy in our results) can be estimated by $p \approx \hat{p} \pm 0.0049$. Therefore, we believe our results are statistically meaningful.
>
> **Question-4: Your paper doesn't explore fine-tuning or more advanced prompting techniques. Couldn't these potentially improve LLM performance significantly?**
>
> First, kindly note that we study chain-of-thought (CoT) prompting in Section 4.3. Our results on the difficult APIs show that while CoT prompting can indeed boost performance (especially for SOTA LLMs like GPT-4-Turbo), it still struggles to solve more complex arithmetic constraints as difficulty increases.
>
> Second, following reviewer YxwD’s suggestion, we evaluate the ReACT prompting strategy in Figure 12 (see attached PDF in global response). We observe that while ReAct can perform better than CoT, it still fails to solve more complex arithmetic constraints. We also follow reviewer WbTY’s suggestion and include API documentation in prompts (Figure 14). We found that adding documentation cannot always achieve better performance on our tasks.
>
> Furthermore, we totally agree that fine-tuning may significantly improve the accuracy of LLMs. However the focus of the paper is to evaluate the **implicit** ability for LLMs to model parameter constraints from pre-training. We hope to evaluate fine-tuning as future work.
>
> **Question-5: Wouldn't categorizing common mistakes provide more insights into LLM limitations?**
>
> Thanks for this suggestion! In our paper, we mainly focus our in-depth analysis based on the constraint types (e.g., equality, inequality, arithmetic, and set-related). Regarding categorizing common mistakes made by LLMs, we did have interesting findings, such as:
>
> - **LLMs struggle with uncommon input tensors**: We found that across many APIs and constraints, LLMs struggle when provided with uncommon input tensor ranks (i.e., rank > 4) or uncommon shapes (e.g., `x = torch.rand(9, 30, 23, 4)`). The reason is that LLMs are mostly trained with data that contains very common shapes or ranks. As such, LLMs can easily make mistakes on uncommon inputs.
>
> - **LLMs tend to predict common parameter values blindly**:  We also observe that LLMs tend to generate common parameter values (e.g., 0, 1, powers of 2) which often turn out to be incorrect. This is again because LLMs are trained with pre-training code that frequently contains such parameter patterns and thus are likely to predict them even given a different input context.
>
> Thanks again for all the great suggestions and we will work towards adding them in the next version of the paper!
>
> **References**
>
> [a] https://github.com/pytorch/pytorch/network/dependents
>
> [b] https://pypi.org/project/numpy/
>
> [c] L2ceval: Evaluating language-to-code generation capabilities of large language models. https://arxiv.org/abs/2309.17446 (2023).

---

> ### Author Response · Authors · 2024-08-13
> **Thanks for your review and great suggestions!**
>
> Thanks again for your great suggestions to categorize and analyze the common error patterns that help us strengthen the paper! We would like to share with you our new interesting findings regarding other common mistakes made by LLMs. In additional to 1) struggling with uncommon input tensors; and 2) predicting common parameter values blindly, we found that 3) **LLMs pay attention to the wrong tokens/irrelevant parameters**: LLMs can learn spurious correlations and pay attention to the wrong context tokens. For example, open-source LLMs struggle with the simple equality constraint `in_features=input.shape[-1]` in `torch.nn.Linear` because the attention weights are focused on the irrelevant parameters.
>
> **Attention analysis:** To give an example, for the DeepSeek-1.3B model and the API `torch.nn.Linear` with difficulty of rank=5, the input tensor has 5 dimensions and is presented in context like `x = torch.randn(5, 4, 2, 14, 11)`, and the model is asked to predict a parameter `in_features`, with the constraint being `in_features==input.shape[-1]`. For this simple task, the LLM only achieves 6% accuracy. To investigate why, we compute the attention weight of each input token (maximum of all attention heads across all layers) to locate the most significant token among the relevant ones (e.g., `5, 4, 2, 14, 11` in the previous example). Next, we map it to a specific dimension of the input tensor (i.e., `0, 1, 2, 3, 4`). Here’s the result:
>
> Out of the 200 tests, here is the number of times that the predicted value matches each input dimension: `{0: 30, 1: 113, 2: 40, 3: 19, 4: 12}`. The correct answer should be `4`, the last dimension of a rank-5 tensor. Note that the sum of this is not 200 because the predicted value can match more than one input dimensions if they are identical.
>
> We observe that the predicted value almost always (98%) matches with the context token that has the highest attention weight. This indicates that the LLM does learn to always copy a specific dimension from the input tensor. As for the detailed break-down: (1) When the attention is paid to the wrong context token/parameter value (92%), it leads to incorrect results; (2) For 1% of the times, it pays attention to the right dimension; (3) Interestingly, for 5% it copies from the wrong position but that specific value just happens to be correct.
>
> We will incorporate your suggestion of including comprehensive error pattern categorization, the attention visualization results, and other remarks in your review in our next revision. Thanks again for your support and valuable comments!

---

### Official Review · Reviewer_qmKc · 2024-07-12

**Soundness:** 2
**Presentation:** 2
**Contribution:** 2
**Rating:** 4
**Confidence:** 5

**Summary:**

This paper constructs a benchmark, DSEval, of 19600 programs across 28 data science (DS) library APIs with numerical constraints and uses the benchmark to invesitigate the capability of LLMs in generating valid DS programs which satisfy those numerical constraints. Additionaly, this paper categories the constraints into four different groups: equality, inequality, arithmetic and set-related. The experiments include 3 different generating settings: 1) full program, 2) all parameters, and 3) individual parameters, and study 8 LLMs including both closed-source and open-source models. The paper shows that LLMs are great at generating simple DS programs, but the performance of LLM drops significantly when the difficulty increases.

**Strengths:**

+ The paper targets at an important problem.
+ The paper is easy to follow in most places.

**Weaknesses:**

- The paper lacks important details.
- Some claims need better justification.
- The experiment is limited in scale in terms of the number of studied APIs.

One contribution of the paper is constructing the benchmark and the paper claims that the design of the benchmark is general and can be easily extended to additional libraries. However, the paper lacks details of how the benchmark is constructed, making it difficult to evaluate this claim. First, lines 11/63/165 state the benchmark contains 28 APIs while line 80 says 12 APIs. The authors need to clarify on this. Second, how the 28 or 12 APIs are selected? what does it mean by representative? Third, do these constraints exist? If so, were they written by developers? What format do these constraints take? Are they in natural language or a formal format? If they are in natural language, how are they converted into a formal format? Forth, are these 19600 programs from existing projects? Fifth, the number of such APIs is large, but the paper only studies 28 or 12 APIs which is pretty a small scale.


Regarding the full program setting, how the 3-step instuction is contructed for individual program? From the example, I can see 3 lines of instructions and 3 lines of code. Are all the programs composed with 3 lines of code? I believe it should not be the case. If there are more than 3 lines of code, does it mean that there will be more than one line of code for each instruction? The paper uses SMT solver to validate the correctness of the generated program. It checks the validation based on constraits. Since the constraints for sepecific for one API. I believe SMT solver can verify the correctness of individul statement. How about the logic of the whole piece of the generated program? It would be better to clarify on this.

**Questions:**

1. Why only 28 or 12 APIs are studied given a large number of APIs?

2. Could you elaborate on the details of the API constraints used in constructing the benchmark? Please see the detailed comments for clarification points.

3. How does the proposed work verify the correctness of the logic of a full program?

**Limitations:**

The paper provides a discussion on limitation.

---

> ### Author Rebuttal · Authors · 2024-08-07
>
> **Question-1: Why only 28 or 12 APIs are studied given a large number of APIs? What does it mean by representative?**
>
> Good question! Please note that although there are a large number of APIs, the commonly used ones (e.g., `Conv2d`) are not that many. For example, the widely-used benchmark DS-1000 [29] contains 1000 DS problems collected from StackOverflow problems (including 68 PyTorch problems and 220 NumPy problems), reflecting realistic use cases; meanwhile, it only covers 16 PyTorch APIs and 59 NumPy APIs (after excluding data construction APIs like `np.ones`). NNSmith [32], a popular DNN generator for testing ML libraries, chooses to support only 73 core operators that are commonly used in DL programs. Furthermore, not all of the commonly used APIs have numeric parameter constraints.
>
> As for our API selection process, first, we followed prior work NNSmith and examined all 73 core operators it supports. Next, we select the 22 core APIs that have numeric parameter constraints and add additional 6 APIs to obtain the 28 APIs used in our study in the full program prediction setting (Section 4.1) and the full API parameter prediction setting (Section 4.2). Furthermore, we choose 12 APIs to cover the representative types of numeric constraint for detailed analysis in the single API parameter prediction setting (Section 4.3) and in our DSeval benchmark (Section 4.4). We use “representative” to mean representative with respect to the numeric parameter constraints in DS library APIs. Table 1 shows the categorization of the different types of numeric constraints that exist in DS libraries. Our selection criteria aim to select a list of APIs that have interesting numeric parameter constraints that can cover all the major constraint categories. You can also find a complete list of the 12 APIs and their corresponding constraints in Table 3 in the Appendix.
>
> **Question-2: Could you elaborate on the details of the API constraints used in constructing the benchmark?**
>
> In this work, we focus on the numeric constraints that are part of the popular DS APIs. These constraints are directly embedded into each individual API. In other words, to use these APIs, the generated DS code must satisfy these constraints. Please refer to Figure 1 for an example of a DS API and its corresponding constraint.
>
> The constraints are defined by developers according to the functionality of each DS API. These constraints are usually specified in natural language within the API documentation. In our study, we manually encode these constraints as satisfiability modulo theory (SMT) formulas and use an SMT solver (Z3) to check if the parameter values generated by the LLM are correct. Please see Section 2.3 for more detail.
>
> The 19,600 programs in our DSeval benchmark are randomly generated by us and are not taken from existing projects. Each benchmark problem requires the LLM to generate a single parameter for an API in order to produce a valid program. To create these problems, we randomly generate values for both the input data and the other parameters in the target API. Using the encoded constraints, we ensure the problem is valid (i.e., the constraints are satisfiable) and retry if it cannot be satisfied. Again, please see Section 2.3 for more detail on the input generation process.
>
> **Question-3. Regarding the full program setting, ... Are all the programs composed with 3 lines of code? I believe it should not be the case. ... How does the proposed work verify the correctness of the logic of a full program?**
>
> Thanks for asking the question, and we believe there may be some misunderstanding. Please note that this paper focuses on ***simple DS programs with only a single API call*** in all settings, including the ***Full program*** setting (see Line-109: “For the full program setting, we want the LLM to synthesize a complete DS program using a specific API from scratch”). We present the full prompt in Figure 2. While LLM-generated programs may contain arbitrary lines of code (e.g., sometimes there are more computations after calling the target API), we extract and focus on the first few code statements only (i.e., the input data generation statements followed by a single API invocation statement). We apologize for any confusion and will revise our presentation for clarity.
>
> ***Rationale of studying a single API***: First, isolating the evaluation to individual APIs or individual API parameters makes it easier to analyze the result. Such a fine-grained setting facilitates a detailed examination of the LLMs’ limitations with respect to various types of numerical constraint. Second, given that LLMs already struggle with constraints within a single API, we believe that expanding the benchmark to multiple APIs would likely yield accuracy too low and difficult to interpret meaningfully. We appreciate the reviewer's comments and will include a discussion of this future work in our revised manuscript.
>
> ***Extend to multiple APIs***: Please note that our method can easily be extended to more complex programs, which is essentially a computation graph consisting of multiple operators and their connections. Since we already symbolically model each individual operator (including input, output, constraints, type transfers), we can also combine these operators and symbolically generate and validate a full computation graph. To achieve this, we can reuse and modify the generation and validation framework provided by NNSmith [32], a popular tool which generates diverse DL computation graphs via formal constraint solving. For more details, please refer to the NNSmith paper.

---

> > ### Comment · Area_Chair_CKy5 · 2024-08-13
> >
> > From Area Chair:  This reply is from reviewer qmKc.  I believe it was accidentally posted without the "visible to authors" box checked.  :-)
> >
> > ---
> > Thank you for the rebuttal. I would keep my ratings after reading the rebuttal and some thinking. My main concern is the number of PIS and complexity of the problem. Compared with similar benchmarks in such kind, the evaluation is small and not comprehensive. The rebuttal mentions that this can be extend without real evidence.

---

> ### Author Response · Authors · 2024-08-13
>
> To Area Chair: Thanks for noticing and posting the comment!
>
> To Reviewer qmKc:
>
> Thanks for reading our rebuttal and for your reply!
>
> > My main concern is the number of PIS and complexity of the problem. Compared with similar benchmarks in such kind, the evaluation is small and not comprehensive.
>
>
> We kindly point out here that DSeval is the **first** benchmark targeting the validity of DS API parameter constraints, consisting of **19600** different problems that span **fine-grained** settings and difficulty levels for 12 APIs, and we evaluated across 8 state-of-the-art open/close source code LLMs. For comparison, DS-1000 [29], which is the closest benchmark we found in the DS code generation domain, contains 1000 problems. Would you kindly share examples of “similar benchmarks in such kind” you have in mind?
>
> Please allow us to further clarify our criteria for dataset construction.
>
> Firstly, our chosen APIs are the core APIs commonly used by users. While there are a large number of APIs in DS libraries, the commonly used ones (e.g., Conv2d) are not that many. For example, the widely-used benchmark DS-1000 [29] only covers 16 PyTorch APIs and 59 NumPy APIs (after excluding data construction APIs like “np.ones”).
>
> Additionally, not all the APIs have numerical parameter constraints. For example, NNSmith [32] supports 73 core operators that are commonly used in DL programs, and only 22 of them have numerical constraints that fall in the scope of our study, and we’ve included *all 22* of them (plus 6 additional ones) in our first two settings (full program generation and all parameter generation).
>
> Furthermore, some APIs have similar constraints or same constraint types. For example, numpy.max and numpy.min have the same constraints, whereas torch.nn.Conv2d and torch.nn.Conv1d have very similar constraints. Therefore, in our DSeval benchmark, we select 12 representative APIs to keep the experiments at an affordable scale, while still ***covering all major constraint categories*** in Table 1. We believe this is a representative and **comprehensive (in terms of constraint types)** set of APIs and constraints used to evaluate LLMs’ capabilities.
>
> > The rebuttal mentions that this can be extended without real evidence.
>
> Thanks for asking this follow-up question. We apologize that we didn't include concrete evidence in our initial response, as it wasn't specifically requested at that time. Please find the supporting evidence below:
>
> We can extend the experiment to new APIs with little engineering effort. For example, we’ve already added torch.nn.Linear into our benchmark during the rebuttal as suggested by reviewer bNWw, and obtained interesting results. Building upon the original framework, we only need to add 17 lines of code for the newly supported API. We didn’t include more APIs due to computing budget constraints, but we are happy to extend the study to all 28 APIs and even more if the reviewer thinks it is crucial.
>
> As we also mentioned in our previous response, the reason that we can easily support the addition of new APIs is because our framework uses symbolic constraint solving techniques to both generate and validate them. New APIs can be added by simply encoding the numeric constraints (written as basic z3 formulas). Similarly, we can also add more complex chained API sequences using our framework by again symbolically modeling each API to build the symbolic computation graph and construct a program. Please note that we built our framework upon NNSmith [32], and thus such graph-level modeling is intrinsically supported and only requires minimal modification to support our use case.
>
> More concretely, **to verify the correctness of the logic of a full program, we just need to symbolically build the API graph, propagate the tensor shapes, and verify the validity of each API.** To give a more concrete example, let us consider two APIs with *symbolic shapes and symbolic parameters*:
>
> - F: Input shape [x,y], parameter a; Output shape [x, a]; Constraint: a<y
> - G: Input shape [x,y], parameter b; Output shape [x, y-b]; Constraint: y>b
>
> Chaining F(G(input)) with symbolic input [u,v]:
> 1. G([u,v]) -> [u,v-b]; Constraint: v>b
> 2. F([u,v-b]) -> [u,a]; Constraint: a<v-b
>
> Note that we have already symbolically modeled the shape transfer rules and validity constraints of each individual API. Therefore, the whole process uses symbolic expressions, allowing seamless propagation and validity checking through the API chain without modifying individual APIs.
>
> Please let us know if our response has addressed your concern, and we are more than happy to provide further evidence if the reviewer is willing to specify which aspects they’d like us to elaborate on or what particular evidence they’re seeking.

---

### Author Rebuttal · Authors · 2024-08-07

We thank all the reviewers for their insightful comments and suggestions to improve the paper! We address the main questions and concerns in this rebuttal. Furthermore, we also plan to revise the paper accordingly to address all other minor suggestions and comments.

We have also attached a PDF corresponding to the new experimental results requested by the reviewer, please kindly see the attached PDF for the detailed result figures.

Please kindly let us know if there is any misunderstanding of the questions, and we are very happy to provide further updates or clarifications during the reviewer-author discussion period.

---

### Decision · Program_Chairs · 2024-09-25

**Decision:**

Accept (poster)

**Comment:**

The authors create a new data set called DS_EVAL, which tests LLMs ability to reason about numeric constraints when doing code generation. The data set consists of 19,600 programs, which call 28 different APIs from pytorch and numpy, and each call is ranked in terms of difficulty.  Constraints refer to things like specifying valid dimension numbers given the shape of arrays, specifying valid argument to reshape, specifying valid permutations when doing transpose, etc.   An SMT solver is used to validate that the LLM output satisfies the constraints.

The authors compare a number of LLMs, and find that LLMs are generally able to handle easy cases using common shapes that occur frequently in training data, but cannot reason about constraints in harder and less common cases.

Reviewer responses to this paper were mixed.  Three reviewers recommend "weak accept" (6), one recommends "weak reject" (4), and the last recommends "reject" (3).   Praise for the paper was quite uniform:

"Originality: First systematic study of LLMs' ability to handle numeric constraints" (reviewer xaud)
"first, thorough and systematic study of these constraints"  (reviewer bNWw)
"The paper explores a new and interesting problem" and "The study is thoroughly conducted"  (reviewer WbTY)

Reviewer (qmKc) [weak reject] is concerned that "the paper lacks important details." -- namely how the constraints were determined and formalized, and how the example programs are constructed.  They also complain that "the experiment is limited in scale in terms of the number of studied APIs."  IMHO, these are valid criticisms, but are not strong enough to reject the paper.

Reviewer (YxwD) [reject] complains that the "paper just provides a comprehensive evaluation on existing LLMs, without proposing any possible solution for this problem" and suggests that the paper should have been submitted to the datasets and benchmarks track instead.

Overall, I personally feel that this is a valuable paper, which is worthy of publication.  The paper examines a particular aspect of code generation that has not been previously explored in the literature, and is likely to be increasingly important as LLMs are applied to more complex coding tasks.  It provides a comprehensive evaluation of existing SOTA models.  Moreover, because this is a benchmark, it can be used by other researchers to test and improve LLM coding capabilities going forward.

However, I do think that reviewer (YxwD) has a point.  This is clearly a "datasets and benchmarks" paper, and perhaps should have been submitted to that track.